# LD3M: Latent Dataset Distillation with Diffusion Models

## Abstract

The concept of dataset distillation, which condenses large datasets into smaller but highly representative synthetic samples, is gaining significant traction because it addresses some of modern AI's core challenges, such as preserving the privacy of training data or storing replay memory samples for continual learning. However, unlocking the full potential of dataset distillation remains difficult due to two main issues. The first is architecture generalization: the distilled dataset often performs well with the architecture used during distillation, typically ConvNet, but struggles to generalize to others. The second is effectively distilling images at resolutions commonly found in standard datasets, such as 128x128 and 256x256. This paper introduces Latent Dataset Distillation with Diffusion Models (LD3M), a novel approach that is the first to combine a modified diffusion process in latent space with dataset distillation to address these issues. LD3M allows for more effective distillation of images with resolutions of $128 \times 128$ or $256 \times 256$ and improved generalization across various architectures. Additionally, LD3M enables fine control over both distillation speed and dataset quality by adjusting the number of diffusion steps. Experimental results demonstrate that LD3M outperforms state-of-the-art methods by up to 4.8 percentage points for one image per class and 4.2 percentage points for ten images per class across several ImageNet subsets.

## 1 Introduction

The undeniable success of employing larger datasets has led to a question captivating a growing body of research (Cazenavette et al., 2023; Ramesh et al., 2022; Bengio et al., 2019): Are large-scale datasets necessary to solve complex visual tasks? Using such datasets demands special equipment and infrastructure. Further, it poses challenges regarding storing and data pre-processing (Ganguli et al., 2022). As an alternative, dataset distillation focuses on generating a small set of representative synthetic samples from large datasets, often condensing them into as few as one image per class (Wang et al., 2018). This approach has gained significant traction due to its versatility, particularly in applications such as privacy preservation (Dong et al., 2022) - where sensitive training data can be substituted with abstract yet representative synthetic data - and accelerated neural architecture search (Such et al., 2020; Masarczyk & Tautkute, 2020). Nevertheless, distillation methods like Dataset Condensation (Zhao et al., 2020), Distribution Matching (Zhao & Bilen, 2023), and Matching Training Trajectories (Cazenavette et al., 2022) still face major limitations. The fundamental problems are the generalization across architectures and the handling of image resolutions commonly found in standard datasets, such as $128 \times 128$ or $256 \times 256$. Classifiers show peak performance when their architecture matches the one used for distillation. However, performance tends to deteriorate when other architectures are trained on the distilled data. A related challenge is that dataset distillation struggles with images larger than $32 \times 32$, primarily because the common practice of distilling raw pixel values results in overfitting to the specific architecture used during distillation (Baradad Jurjo et al., 2021; Cazenavette et al., 2023).

A promising solution is to integrate a generative prior into dataset distillation, as generative models have already demonstrated remarkable success in producing high-fidelity images at resolutions beyond $32 \times 32$ (Frolov et al., 2024; Bar-Tal et al., 2023). By leveraging the capabilities of generative models, it becomes possible to distill synthetic latent codes that can be decoded to synthetic images that generalize better across architectures and perform well at higher resolutions. One notable example is GLaD (Cazenavette et al., 2023), which combines StyleGAN-XL with dataset distillation algorithms.

Despite this innovative approach, GLaD's performance still has much room for improvement. A key limitation of GLaD lies in its reliance on GANs, which, while effective, do not fully exploit the advantages offered by more recent and superior generative models, such as diffusion models. However, incorporating diffusion models into dataset distillation introduces its own set of challenges. One of the most significant issues is the vanishing gradient problem that can occur during distillation due to the many iterative time steps in the diffusion process (Hochreiter, 1998). This makes maintaining a strong gradient flow throughout the distillation procedure difficult, especially when dealing with images of resolution $128 \times 128$ or $256 \times 256$. To overcome these challenges, it is crucial to design a distillation process that can effectively balance the strengths of diffusion models while mitigating the gradient vanishing problem.

This is where Latent Dataset Distillation with Diffusion Models (LD3M) comes in. Our work contrasts recent attempts to integrate diffusion in the distillation process like Su et al. (2024) and Duan et al. (2023), which primarily utilize the autoencoder encapsulating the diffusion model to avoid the vanishing gradient problem: Instead of relying on averaged encoded codes or solely utilizing the decoder, LD3M enables for the first time the effective learning of latent codes through the entire diffusion process. This is achieved by introducing a modified diffusion process that enhances gradient flow, enabling more effective distillation of high-resolution images while ensuring that the distilled datasets generalize well across diverse model architectures.

Another benefit is that LD3M leverages pre-trained diffusion models, enabling it to perform efficient dataset distillation without fine-tuning the diffusion model. Moreover, the flexible framework of LD3M supports any diffusion model, making it highly extensible and adaptable to emerging models. Thus, new and pre-trained diffusion models can be easily integrated. Finally, our experiments show that LD3M distills notably faster and generates $128 \times 128$ and $256 \times 256$ datasets of much higher quality for training various architectures than the state-of-the-art latent dataset distillation method GLaD. Overall, our work:

- presents LD3M as the first latent distillation method that fully leverages the diffusion process. In contrast, others rely solely on pre-trained decoders or averaging encoded codes, limiting their ability to truly learn representations.
- introduces a novel diffusion process, designed specifically for dataset distillation. By enhancing gradient flow through residual connections, LD3M not only accelerates the distillation process but also sets a new foundation for future innovations in generative modeling.
- offers a seamless integration into existing distillation frameworks without requiring further training or customization.
- achieves superior results across multiple benchmarks, significantly outperforming the state-of-the-art method GLaD. LD3M consistently generates more diverse and expressive synthetic datasets, demonstrating its ability to improve training efficiency and generalization across unseen architectures.

## 2 Background

Recent studies have shown that many large-scale datasets include redundant samples that do not affect model training and can be ignored (Katharopoulos & Fleuret, 2018). This insight inspired methods that try to decrease the size of training sets, *e.g.*, importance sampling and coreset selection (Moser et al., 2022; Guo et al., 2022; Mirzasoleiman et al., 2020; Csiba & Richtárik, 2018). In a similar spirit, dataset distillation methods have emerged to address the inefficiencies of large datasets, such as frameworks like SRe2L (Yin et al., 2024), which focus on decoupling synthesis processes to reduce the computational cost of traditional distillation. In this Section, however, we will only review the foundational algorithms in dataset distillation that are relevant to our analysis and discuss their relation to LD3M.

### 2.1 Dataset Distillation

Let $\mathcal{T} = (X_r, Y_r)$, where $X_r \in \mathbb{R}^{N \times H \times W \times C}$, be a real image classification dataset and $N$ its number of samples. The goal is to compress $\mathcal{T}$ into a small synthetic set $\mathcal{S} = (X_s, Y_s)$, where $X_s \in \mathbb{R}^{M \times H \times W \times C}$, where $M$ is the total number of synthetic samples with $M = \mathcal{C} \cdot IPC$, $\mathcal{C}$ the number of classes and $IPC$ the Images Per Class (IPC). We aim to achieve $M \ll N$, and describe the problem as:

$$\mathcal{S}^* = \arg\min_{\mathcal{S}} \mathcal{L}(\mathcal{S}, \mathcal{T}), \tag{1}$$

where $\mathcal{L}$ is some objective for dataset distillation, which will be discussed in the next Sections. Broadly speaking, the synthetic images are learnable, similar to weights in neural networks. After distillation, each synthetic image constitutes an expressive representation of a class in the original dataset. Most approaches initialize the synthetic images with IPC-many random images from each class. In the following, we describe three commonly used methods to define $\mathcal{L}$, namely Dataset Condensation (Zhao et al., 2020), Distribution Matching (Zhao & Bilen, 2023), and Matching Training Trajectories (Cazenavette et al., 2022).

**Dataset Condensation (DC)** ensures alignment by deriving the gradients via a classification error (Zhao et al., 2020). It calculates the loss on real ($\ell^{\mathcal{T}}$) and the respective synthetic data ($\ell^{\mathcal{S}}$). Next, it minimizes the distance between the gradients of both network instances. More concretely,

$$\mathcal{L}_{DC} = 1 - \frac{\nabla_\theta \ell^{\mathcal{S}}(\theta) \cdot \nabla_\theta \ell^{\mathcal{T}}(\theta)}{\|\nabla_\theta \ell^{\mathcal{S}}(\theta)\| \, \|\nabla_\theta \ell^{\mathcal{T}}(\theta)\|}. \tag{2}$$

**Distribution Matching (DM)** obtains gradients by minimizing the logits on the real and synthetic datasets. It enforces the feature extractor (ConvNet) to produce similar features for real and synthetic images (Zhao & Bilen, 2023). The distribution matching loss is

$$\mathcal{L}_{DM} = \sum_c \left\| \frac{1}{|\mathcal{T}_c|} \sum_{\mathbf{x} \in \mathcal{T}_c} \psi(\mathbf{x}) - \frac{1}{|\mathcal{S}_c|} \sum_{\mathbf{s} \in \mathcal{S}_c} \psi(\mathbf{s}) \right\|^2, \tag{3}$$

where $\mathcal{T}_c, \mathcal{S}_c$ are the real and synthetic images for a class $c$.

**Matching Training Trajectories (MTT)** does not rely on the gradients obtained by calculating the classification error or the feature distance. Instead, it leverages the network parameters, *i.e.*, ConvNet, (Cazenavette et al., 2022). In more detail, this approach exploits several trained instances of the model on the original dataset, called experts, and stores the training trajectory of parameters $\{\theta_t^*\}_0^T$ at predetermined intervals, called expert trajectories. For dataset distillation, MTT samples a random set of parameters $\theta_t^*$ from the trajectory at a given timestamp. Next, it trains a new network, $\hat{\theta}_{t+N}$, initialized with the parameters on the respective synthetic images (for $N$ iterations). Finally, the distance between the trajectory on the real dataset, $\theta_{t+M}^*$ with $M$ steps, and the trajectory on the synthetic one, $\hat{\theta}_{t+N}$, is minimized. As a result, MTT tries to mimic the original dataset's training path (trajectory of parameters) with the synthetic images:

$$\mathcal{L}_{MTT} = \frac{\|\hat{\theta}_{t+N} - \theta_{t+M}^*\|^2}{\|\theta_t^* - \theta_{t+M}^*\|^2}. \tag{4}$$

## 2.2 Dataset Distillation with Generative Prior

Dataset distillation with a deep generative model as the generative prior optimizes the latent codes of a pre-trained generative model instead of directly focusing on the raw pixel values (Cazenavette et al., 2023). More formally, let $\mathcal{D} : \mathbb{R}^{M \times h \times w \times d} \to \mathbb{R}^{M \times H \times W \times C}$ be a generator or decoder and $h \cdot w \cdot d \ll H \cdot W \cdot C$. As a result, we can reformulate Equation 1 to:

$$\mathcal{Z}^* = \arg\min_{\mathcal{Z}} \mathcal{L}(\mathcal{D}(\mathcal{Z}), \mathcal{T}) \tag{5}$$

$\mathcal{L}$ is again some objective for dataset distillation, thereby agnostic to the applied distillation algorithm. One of the first methods exploiting generative priors is GLaD (Cazenavette et al., 2023), which uses a pre-trained StyleGAN-XL (Sauer et al., 2022). Yet, StyleGAN-XL offers a wide variety of latent spaces for generating images. This requires evaluating all possible latent spaces with various distillation methods to find the most suitable one, which is a time-consuming process. Moreover, even distilling into the more expansive $\mathbf{W}^+$ latent space is overly restrictive, as it limits synthetic samples to resemble images from the training dataset of StyleGAN-XL (Abdal et al., 2019). But, as synthetic samples compress a whole set of images, they do not need to look real. Concurrent works, such as Zhong et al. (2024), try to mitigate this problem by proposing an extensive greedy search spanning different hierarchical latent spaces of StyleGAN-XL. In contrast, LD3M offers a single latent space efficient enough to express synthetic images of higher quality than GLaD.

Another drawback of using GANs is that they restrict the common practice of initializing synthetic samples with real images (Kim et al., 2022; Liu et al., 2023; Zhao & Bilen, 2021). That would mean we must derive the latent codes leading to the real images by applying $\mathcal{D}$. However, getting the latent codes of an image for a GAN is similar to a pseudo-inversion task, connecting to the broader research domain of GAN inversion (Brock et al., 2017; Zhu et al., 2016). The central concept involves mapping an actual input image into GAN latent codes, where selecting a particular latent code balances expressiveness and reconstruction fidelity, as evidenced in prior research (Tov et al., 2021; Zhu et al., 2020). The cost-intensive GAN inversion process is a significant drawback compared to our method LD3M, as LD3M can initialize the initial latent codes by straightforwardly applying a pre-trained encoder $\mathcal{E}$.

### 2.3 Diffusion Models for Image Generation

Diffusion models are a class of generative models that learn to generate data by gradually transforming simple noise into complex, structured outputs. They achieve this by modeling the process in two phases: a forward process, $q$, where noise is added to the data, and a reverse process, $p$, where the noise is removed to recover the original data distribution. This reversal aids in approximating a complex target data distribution gradually (Yang et al., 2023; Moser et al., 2023c). The main characteristic of diffusion models, which sets them apart from previous generative models, is their dependency on previous time steps. Through iterative refinement, the generative model keeps track of small perturbations and corrects them instead of predicting a large and challenging transformation like GANs (Ho et al., 2020; Nichol & Dhariwal, 2021).

To reduce the computational requirements for generating high-resolution images, Rombach et al. (2022) propose to move the diffusion process towards the latent space representations of an autoencoder structure, which they called Latent Diffusion Model (LDM). Usually, a pre-trained autoencoder (Esser et al., 2021), defined by the encoder $\mathcal{E} : \mathbb{R}^{M \times H \times W \times C} \to \mathbb{R}^{M \times h \times w \times d}$ and decoder $\mathcal{D} : \mathbb{R}^{M \times h \times w \times d} \to \mathbb{R}^{M \times H \times W \times C}$, is used to compress input data into a low-dimensional latent code $\mathcal{Z} \in \mathbb{R}^{M \times h \times w \times d}$. Next, LDM uses the diffusion pipeline in the latent space and employs a decoder to translate the result, the processed latent code, denoted by $\mathbf{z}_0$, back to pixel space via $\mathcal{D}(\mathbf{z}_0)$. The exploit of the latent space is essential for LD3M, as we want to optimize the latent codes $\mathcal{Z}$ that generate synthetic images.

## 3 Latent Dataset Distillation with Diffusion Models (LD3M)

LD3M leverages the generative power of diffusion models for dataset distillation. The presented model focuses on generating synthetic images in the latent space to address the generalization problem of unseen architectures and images of higher resolution than 32x32, *i.e.*, 128x128 and 256x256. The following will explain the sampling process, which takes the latent and conditioning code as input and generates the synthetic image. Next, we describe how we initialize the latent codes before distillation and, finally, how we use gradient checkpointing to reduce VRAM consumption. Figure 1 shows a general overview of LD3M.

### 3.1 Sampling Process

Since we use a pre-trained LDM without fine-tuning, we can primarily focus on the backward diffusion process $p$. It starts from an initial state $\mathbf{z}_T$, usually Gaussian noise, and performs the inference conditioned on $\mathbf{c}$, *i.e.*, an embedding of the class label (Yang et al., 2023; Moser et al., 2024). The forward diffusion process $q$, on the other hand, is only needed to derive $\mathbf{z}_T \sim q(\mathbf{z}_T \mid \mathcal{Z})$ from the learned and distilled latent representations $\mathcal{Z}$. Therefore, we can view $\mathbf{z}_T$ as the distorted representation of $\mathcal{Z}$, which the LDM refines. We approximate $p$ with a parameterized time-conditional process $p_\theta$, such that

$$p_\theta(\mathbf{z}_{0:T}|\mathbf{c}) = p(\mathbf{z}_T) \prod_{t=1}^{T} p_\theta(\mathbf{z}_{t-1}|\mathbf{z}_t, \mathbf{c}) \tag{6}$$

$$p(\mathbf{z}_T) = \mathcal{N}(\mathbf{z}_T \mid \mathbf{0}, \mathbf{I}) \tag{7}$$

$$p_\theta(\mathbf{z}_{t-1}|\mathbf{z}_t, \mathbf{c}) = \mathcal{N}(\mathbf{z}_{t-1} \mid \mu_\theta(\mathbf{c}, \mathbf{z}_t, \gamma_t), \sigma_t^2 \mathbf{I}). \tag{8}$$

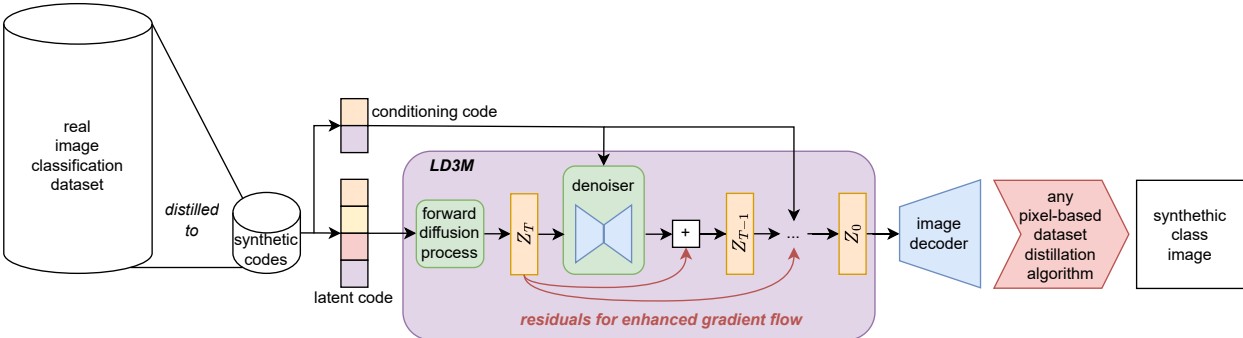

Figure 1: Overview of the LD3M framework. Two components influence the generation of the synthetic images: The distilled latent codes and the distilled conditioning codes. The distilled latent codes are perturbated via Gaussian noise to the initial state $\mathbf{z}_T$. Next, it is iteratively denoised $(T-1)$ times with the pre-trained denoising U-Net of the LDM. Within each computation of the intermediate state $\mathbf{z}_t$, we add a linearly decreasing influence of $\mathbf{z}_T$ to allow an enhanced gradient flow to the distilled latent codes while making the conditioning also learnable. The pre-trained decoder translates the final latent code $\mathbf{z}_0$ back to pixel space. Note that LD3M can be used with any existing distillation algorithm, *e.g.*, DC, DM, or MTT.

As a standard procedure in the literature (Li et al., 2022; Saharia et al., 2022), we predict the parameterized mean by subtracting the scaled noise between two subsequent time steps with

$$\mu_\theta(\mathbf{c}, \mathbf{z}_t, \gamma_t) = \frac{1}{\sqrt{\alpha_t}} \left( \mathbf{z}_t - \frac{1-\alpha_t}{\sqrt{1-\gamma_t}} f_\theta\left(\mathbf{c}, \mathbf{z}_t, \gamma_t\right) \right), \tag{9}$$

where $f_\theta\left(\mathbf{c}, \mathbf{z}_t, \gamma_t\right)$ is the noise prediction of $\varepsilon_t$ with a time-conditional U-Net at time step $t$. Together with the variance $\sigma_t^2$, we can calculate the subsequent state $\mathbf{z}_{t-1}$ via

$$\mathbf{z}_{t-1} \leftarrow \mu_\theta(\mathbf{c}, \mathbf{z}_t, \gamma_t) + \sigma_t^2 \varepsilon_t. \tag{10}$$

For dataset distillation, we make the embedded conditioning $\mathbf{c}$ as well as the latent code $\mathcal{Z}$ leading to the initial state $\mathbf{z}_T$ learnable. Since the LDM is not trained from scratch, merely modifying the condition state $\mathbf{c}$ is insufficient for generating expressive synthetic images, necessitating that $\mathcal{Z}$ be learnable (Park et al., 2023; Kwon et al., 2022). We will show this quantitatively in the experiments, which indicate that making only the conditioning $\mathbf{c}$ learnable predominantly produces images resembling real data. Our overall optimization objective for a given class becomes

$$\mathcal{Z}^*, \mathbf{c}^* = \underset{\mathcal{Z}, \mathbf{c}}{\arg\min}\, \mathcal{L}(\mathcal{D}[p_\theta(\mathbf{z}_0|\mathbf{z}_T, \mathbf{c})], \mathcal{T}), \text{ with } \mathbf{z}_T \sim q(\mathbf{z}_T|\mathcal{Z}) \tag{11}$$

A significant challenge in making the latent code $\mathcal{Z}$, which leads to the initial state $\mathbf{z}_T$, learnable arises from the numerous time steps during the backward diffusion process, which can take $T = 200$ steps or more (Rombach et al., 2022). This extensive computation chain leads to vanishing gradients for $\mathbf{z}_T \sim q(\mathbf{z}_T \mid \mathcal{Z})$, impeding an effective distillation of $\mathcal{Z}$ (Hochreiter, 1998). We modify Equation 10 to counteract this by including residual connections in the computational graph, thereby enhancing gradient flow that work reliably to timesteps up to $T = 40$, which we will show empirically for $T = 10$ and $T = 20$ in our experiments. Specifically, we integrate the initial state $\mathbf{z}_T$ into the computation of the intermediate states $\mathbf{z}_t$. Additionally, we systematically diminish its influence as $t$ approaches 0. This adjustment ensures an enhanced gradient propagation crucial for the generation of diverse and representative synthetic latent codes $\mathcal{Z}$:

$$\mathbf{z}_{t-1} \leftarrow \left( (1 - \frac{t}{T}) \cdot \mu_\theta(\mathbf{c}, \mathbf{z}_t, \gamma_t) + \frac{t}{T} \cdot \mathbf{z}_T \right) + \sigma_t^2 \varepsilon_t. \tag{12}$$

This formulation leads to the following enhanced gradient flow with the initial latent code $\mathcal{Z}$ at $t = T$:

$$\frac{\partial \mathcal{L}}{\partial \mathcal{Z}} = \sum_{t=1}^T \left( 1 - \frac{T-1}{T} \right) \cdot \left[ \frac{\partial \mathcal{L}}{\partial \mathbf{z}_t} \cdot \frac{\partial \mathbf{z}_t}{\partial \mathbf{z}_{t-1}} \cdot \ldots \cdot \frac{\partial \mathbf{z}_0}{\partial \mathcal{Z}} \right] + \left( \frac{t}{T} \right) \cdot \left[ \frac{\partial \mathcal{L}}{\partial \mathbf{z}_T} \cdot \frac{\partial \mathbf{z}_T}{\partial \mathcal{Z}} \right] \tag{13}$$

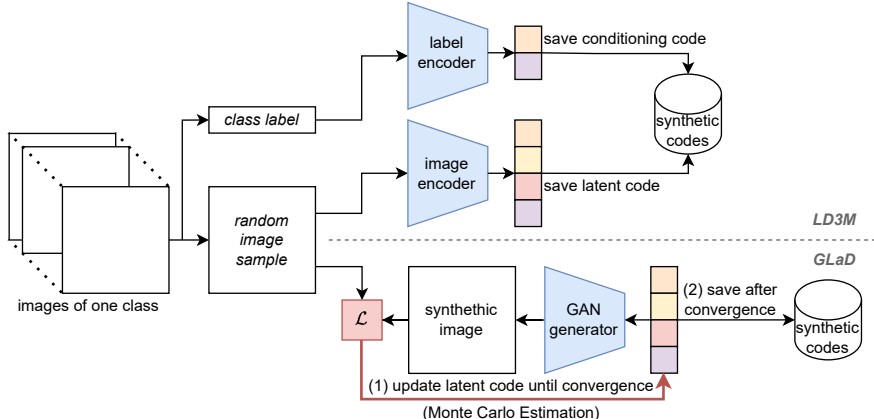

Figure 2: Latent code initialization based on real samples of LD3M and GLaD. While GLaD needs multiple steps to sample a latent code that leads to a synthetic image close to a real image, LD3M applies a pre-trained image encoder. Moreover, LD3M uses a label encoder to save additional information for image generation.

A comprehensive algorithmic description can be found in the appendix.

**Note on Markovian Property:** $\mathbf{z}_{t-1}$ depends on $\mathbf{z}_t$ and $\mathbf{z}_T$, but not on any earlier states such as $\mathbf{z}_{t+1}$, $\mathbf{z}_{t+2}$, and so on. Therefore, the probability distribution for $\mathbf{z}_{t-1}$ only depends on $\mathbf{z}_t$ and the fixed initial state $\mathbf{z}_T$, which is constant throughout the diffusion process. Thus, we have: $p(\mathbf{z}_{t-1}|\mathbf{z}_t, \mathbf{z}_{t+1}, \ldots, \mathbf{z}_T) = p(\mathbf{z}_{t-1}|\mathbf{z}_t, \mathbf{z}_T)$. This confirms that LD3M remains Markovian. Notes on the noise addition are in the appendix (A.1).

**Note on Generalisability:** This reformulation is not bound to LDMs and can be used by any diffusion model type for future work. Nevertheless, we focused on LDM because of several advantages: (a) there exists a pre-trained model, (b) it is foundational for diffusion models (especially in exploiting latent space), and (c) it serves as a proof of concept.

## 3.2 Initializing Latent Codes

As standard in dataset distillation, we want to initialize the synthetic images using random samples from the respective class. With GLaD, which relies on StyleGAN-XL, computationally expensive GAN-inversion techniques are required to obtain the latent code of a real image, making the initialization process far more complex. With GAN-inversion, one has to solve the optimization problem $\mathcal{Z}^* = \arg\min_{\mathcal{Z}} \mathcal{L}(\mathbf{x}, G(\mathcal{Z}))$. A solution can be optimization-based, learning-based, or hybrid-based, all needing careful adjustments and testing (Xia et al., 2022; Cazenavette et al., 2023). Moreover, finding solutions is time-consuming as an iterative sampling of possible $\mathcal{Z}^*$ or extra training is required, as illustrated in Figure 2. In contrast, LD3M can initialize the latent codes straightforwardly by applying the pre-trained encoder $\mathcal{E} : \mathbb{R}^{M \times H \times W \times C} \to \mathbb{R}^{M \times h \times w \times d}$ to the random images, *i.e.*, $\mathcal{Z} = \mathcal{E}(X_s)$ for a randomly selected collection $X_s$. Furthermore, LD3M initializes the embedded condition information by applying the pre-trained embedding network. As a result, initialization with LD3M is more straightforward and computationally more efficient than GLaD and omits careful adjustments before applying dataset distillation.

## 3.3 Gradient Checkpointing

Like GLaD (Cazenavette et al., 2023), we employ gradient checkpointing (Chen et al., 2016) to reduce the VRAM requirements. At each distillation iteration, we generate the synthetic images $\mathcal{S} = \mathcal{D}(\mathcal{Z})$ without gradients. Then, we calculate the distillation loss $\mathcal{L}$ and the gradients for the synthetic images ($\partial\mathcal{L}/\partial\mathcal{S}$). Next, we delete the computation graph and its gradient. To compute $\partial\mathcal{L}/\partial\mathcal{Z}$, we recompute the forward pass through $\mathcal{D}$, $\mathcal{S} = \mathcal{D}(\mathcal{Z})$, to obtain $\partial\mathcal{S}/\partial\mathcal{Z}$. The application of the chain rule delivers $\partial\mathcal{L}/\partial\mathcal{Z} = (\partial\mathcal{L}/\partial\mathcal{S})(\partial\mathcal{S}/\partial\mathcal{Z})$ which is used to update the latent codes and the conditioning.

Since our generative model $\mathcal{D}$ applies multiple diffusion steps, we also employ checkpointing within $\mathcal{D}(\mathcal{Z})$ for each noise prediction $\epsilon_t$ of the U-Net. It saves $\partial\mathbf{z}_t/\partial\mathbf{z}_{t+1}$. The application of the chain rule leads to

$$\partial\mathcal{L}/\partial\mathcal{Z} = (\partial\mathcal{L}/\partial\mathbf{z}_0) \cdot \prod_{t=1}^{T} \frac{\partial\mathbf{z}_{t-1}}{\partial\mathbf{z}_t} \cdot (\partial\mathbf{z}_T/\partial\mathcal{Z}) \tag{14}$$

Note that this formulation does not include the residual connections introduced in Equation 12 for the sake of simplicity. Similarly, the condition information $\mathbf{c}$ is derived by

$$\partial\mathcal{L}/\partial\mathbf{c} = (\partial\mathcal{L}/\partial\mathbf{z}_0) \cdot \prod_{t=1}^{T} \frac{\partial\mathbf{z}_{t-1}}{\partial\mathbf{z}_t} \cdot (\partial\mathbf{z}_T/\partial\mathbf{c}) \tag{15}$$

Even though LD3M needs more checkpoints than GLaD, our experiments will show that LD3M is still faster.

## 4 Experiments

We follow the setup of Cazenavette et al. (2023) and evaluate the cross-architecture performance for IPC=1 (MTT, DC, DM) and IP=10 (DC, DM) with image size $128 \times 128$. Furthermore, we run an evaluation with DC and image size $256 \times 256$ for IPC=1 as well as a low-resolution evaluation on CIFAR-10 ($32 \times 32$). Finally, we visually compare the results with GLaD and analyze the impact of different latent and conditioning code initializations, various diffusion lengths, and different components of LD3M. In all experiments, we maintain consistent hyperparameters to guarantee a fair comparison (more details in the appendix, A.3).

### 4.1 Setup

**Datasets.** We evaluate the accuracy of classifiers trained on synthetic images on 10-class subsets of ImageNet-1k (Deng et al., 2009). The 10-class subsets were derived from previous work, *i.e.*, ImageNet-Birds, ImageNet-Fruits, and ImageNet-Cats (Cazenavette et al., 2022). In addition, we employ two other commonly used subsets, namely ImageNette and ImageWoof (Howard, 2019). Lastly, the remaining used subsets are based on the evaluation performance of a ResNet-50 model (pre-trained on ImageNet)(Cazenavette et al., 2023). These subsets are ImageNet-A, consisting of the top-10 classes; ImageNet-B, consisting of the next 10 and so on for ImageNet-C, ImageNet-D, and ImageNet-E. More details can be found in the appendix.

**Evaluation Protocol.** We first distill synthetic datasets using the designated algorithms and then assess the quality of the datasets by measuring the performance across unseen network architectures. To evaluate a synthetic dataset with a specific architecture, we train a new network from scratch on the distilled dataset and subsequently evaluate its performance on the test set (real images). This entire process is replicated five times, with the report including the mean test accuracy plus or minus one standard deviation.

**Network Architectures.** As in GLaD (Cazenavette et al., 2023) and prior dataset distillation work (Nguyen et al., 2021; Cui et al., 2023; Nguyen et al., 2020), we use the ConvNet-5 and ConvNet-6 architecture to distill the $128 \times 128$ and $256 \times 256$ datasets, respectively (Gidaris & Komodakis, 2018). Similarly, we use AlexNet (Krizhevsky et al., 2012), VGG-11 (Simonyan & Zisserman, 2014), ResNet-18 (He et al., 2016), and a Vision Transformer (Dosovitskiy et al., 2020) for evaluating unseen architectures.

**Latent Diffusion Model.** For the diffusion model, we use the ImageNet pre-trained LDM (Rombach et al., 2022), which guarantees a fair comparison with the ImageNet pre-trained StyleGAN-XL used in GLaD (Cazenavette et al., 2023). We use the pre-trained autoencoder with $2\times$ compression of the LDM without further adjustments to compress into and decompress out of the latent space. We employ for all experiments a max. diffusion time step $T = 10$ for image size $128 \times 128$ and $T = 20$ for $256 \times 256$. Besides this hyper-parameter, no adjustment to the pre-trained LDM was necessary. In addition, we want to highlight that using a pre-existing diffusion model is not a limitation but an advantage, showing that a pre-trained diffusion model can be used out of the box (and empirically more consistent as GLaD *w.r.t.* various experiments).

| Distil. Space | Alg. | ImNet-A | ImNet-B | ImNet-C | ImNet-D | ImNet-E | ImNette | ImWoof | ImNet-Birds | ImNet-Fruits | ImNet-Cats |
|---|---|---|---|---|---|---|---|---|---|---|---|
| pixel space | MTT | 33.4±1.5 | 34.0±3.4 | 31.4±3.4 | 27.7±2.7 | 24.9±1.8 | 24.1±1.8 | 16.0±1.2 | 25.5±3.0 | 18.3±2.3 | 18.7±1.5 |
| | DC | 38.7±4.2 | 38.7±1.0 | 33.3±1.9 | 26.4±1.1 | 27.4±0.9 | 28.2±1.4 | 17.4±1.2 | 28.5±1.4 | 20.4±1.5 | 19.8±0.9 |
| | DM | 27.2±1.2 | 24.4±1.1 | 23.0±1.4 | 18.4±0.7 | 17.7±0.9 | 20.6±0.7 | 14.5±0.9 | 17.8±0.8 | 14.5±1.1 | 14.0±1.1 |
| GLaD | MTT | 39.9±1.2 | 39.4±1.3 | **34.9±1.1** | 30.4±1.5 | 29.0±1.1 | 30.4±1.5 | 17.1±1.1 | 28.2±1.1 | 21.1±1.2 | 19.6±1.2 |
| | DC | 41.8±1.7 | **42.1±1.2** | 35.8±1.4 | 28.0±0.8 | 29.3±1.3 | 31.0±1.6 | 17.8±1.1 | 29.1±1.0 | 22.3±1.6 | 21.2±1.4 |
| | DM | 31.6±1.4 | 31.3±3.9 | 26.9±1.2 | 21.5±1.0 | 20.4±0.8 | 21.9±1.1 | 15.2±0.9 | 18.2±1.0 | 20.4±1.6 | 16.1±0.7 |
| **LD3M** | MTT | **40.9±1.1** | 41.6±1.7 | 34.1±1.7 | **31.5±1.2** | 30.1±1.3 | 32.0±1.3 | **19.9±1.2** | **30.4±1.5** | 21.4±1.1 | **22.1±1.0** |
| | DC | **42.3±1.3** | 42.0±1.1 | **37.1±1.8** | 29.7±1.3 | **31.4±1.1** | **32.9±1.2** | 18.9±0.6 | 30.2±1.4 | 22.6±1.3 | 21.7±0.8 |
| | DM | **35.8±1.1** | **34.1±1.0** | 30.3±1.2 | 24.7±1.0 | 24.5±0.9 | 26.8±1.7 | 18.1±0.7 | 23.0±1.8 | **24.5±1.9** | 17.0±1.1 |

Table 1: Cross-architecture performance with 1 image per class on all ImageNet (128×128) subsets. We used the unseen architectures AlexNet, VGG11, ResNet18, and ViT. The average performance of these models was then evaluated on actual validation sets. Using LDM as a deep generative prior markedly enhanced the ability of all tested methods to generalize across various architectures in all the datasets examined. In 9 out of 10 subsets (marked with blue), LD3M reaches the best overall performance. The best performance within one distillation algorithm is marked in bold in each subset, primarily achieved by LD3M, showing that LD3M improves DC, MTT, and DM by +3.76%, +5.68%, and +16.34%, respectively, on average.

| Distil. Space | Alg. | All | ImNet-A | ImNet-B | ImNet-C | ImNet-D | ImNet-E |
|---|---|---|---|---|---|---|---|
| pixel space | DC | 42.3±3.5 | 52.3±0.7 | 45.1±8.3 | 40.1±7.6 | 36.1±0.4 | 38.1±0.4 |
| | DM | 44.4±0.5 | 52.6±0.4 | 50.6±0.5 | 47.5±0.7 | 35.4±0.4 | 36.0±0.5 |
| GLaD | DC | 45.9±1.0 | 53.1±1.4 | 50.1±0.6 | 48.9±1.1 | 38.9±1.0 | 38.4±0.7 |
| | DM | 45.8±0.6 | 52.8±1.0 | 51.3±0.6 | **49.7±0.4** | 36.4±0.4 | 38.6±0.7 |
| **LD3M** | DC | **47.1±1.2** | **55.2±1.0** | 51.8±1.4 | **49.9±1.3** | **39.5±1.0** | **39.0±1.3** |
| | DM | **47.3±2.1** | **57.0±1.3** | **52.3±1.1** | 48.2±4.9 | **39.5±1.5** | **39.4±1.8** |

Table 2: Cross-architecture performance with 10 images per class on the subsets ImageNet A to E. LD3M (marked with blue) achieved the best performance per subset. The best performance within one distillation algorithm is marked in bold in each subset, primarily achieved by LD3M, with an improvement of +2.52% and +3.46% with DC and DM, respectively.

## 4.2 Results

**Cross-Architecture Evaluation, IPC=1.** Our method LD3M ($T = 10$) consistently surpasses the performance of GLaD on unseen architectures for IPC=1 (see Table 1). Following Cazenavette et al. (2023), we test all ten aforementioned subsets of ImageNet and apply the distillation algorithms MTT, DC, and DM. The enhanced performance of our LD3M is consistent across most subsets, except for slight deviations in MTT on ImageNet-C and DC on ImageNet-B. Notably, these exceptions still closely align with the original performance metrics and fall within the standard deviation range, underlining the robustness of our approach. The enhancement in performance is most pronounced when employing the DM distillation algorithm (with +16.34% improvement on average). LD3M with LDM outperforms its GAN-based GLaD counterpart by approximately 4 p.p. across most subsets. This significant improvement underscores the effectiveness of LDMs in capturing complex class feature distributions in synthetic images more effectively than StyleGAN-XL. In conclusion, LD3M improves the gap between pixel-based and latent-based dataset distillation.

**Cross-Architecture Evaluation, IPC=10.** Further supporting our findings, Table 2 illustrates the superior performance of LD3M ($T = 10$) over GLaD in scenarios with IPC=10 using the distillation algorithms DC and DM on ImageNet A to E. The trend observed is similar to the IPC=1 scenario, with LD3M demonstrating a clear advantage over GLaD by improving the performance of DC and DM by +2.52% and +3.46%, respectively. The exception of DM on ImageNet-C is marginal and still outperforms pixel-based distillation, but also shows that the application of LDM for IPC=10 has the most significant improvements in DM compared to DC, *e.g.*, ca. 4 p.p. improvement on ImageNet-A. LD3M reaches an overall average accuracy of 47.08 % with DC and 47.28 % with DM compared to 45.88 % and 45.76 % of GLaD. This experiment highlights the consistency and reliability of the LDM approach across different settings.

| Distil. Space | All | ImNet-A | ImNet-B | ImNet-C | ImNet-D | ImNet-E |
|---|---|---|---|---|---|---|
| pixel space | 29.5±3.1 | 38.3±4.7 | 32.8±4.1 | 27.6±3.3 | 25.5±1.2 | 23.5±2.4 |
| GLaD (ImageNet) | 34.4±2.6 | 37.4±5.5 | 41.5±1.2 | 35.7±4.0 | 27.9±1.0 | 29.3±1.2 |
| GLaD (Random) | 34.5±1.6 | 39.3±2.0 | 40.3±1.7 | 35.0±1.7 | 27.9±1.4 | 29.8±1.4 |
| GLaD (FFHQ) | 34.0±2.1 | 38.3±5.2 | 40.2±1.1 | 34.9±1.1 | 27.2±0.9 | 29.4±2.1 |
| **LD3M** (ImageNet) | 36.3±1.6 | **42.1±2.2** | **42.1±1.5** | 35.7±1.7 | 30.5±1.4 | 30.9±1.2 |
| **LD3M** (Random) | **36.5±1.6** | 42.0±2.0 | 41.9±1.7 | **37.1±1.4** | 30.5±1.5 | **31.1±1.4** |
| **LD3M** (FFHQ) | 36.3±1.5 | 42.0±1.6 | 41.9±1.6 | 36.5±2.2 | **30.5±0.9** | 30.6±1.1 |

Table 3: 256×256 distilled HR images using the DC distillation algorithm and IPC=1. For both scenarios (LDM and GAN), we evaluate pre-trained generators on ImageNet (Deng et al., 2009), FFHQ (Karras et al., 2019), and randomly initialized. Again, LD3M improves the performance of GLaD by roughly +6.03% on average, whereas it improves the performance by roughly +23.28% compared to pixel space.

| Dist. Method | Dist. Space | AlexNet | ResNet18 | VGG11 | ViT | Average |
|---|---|---|---|---|---|---|
| DC | pixel space | 25.9±0.2 | 27.3±0.5 | 28.0±0.5 | 22.9±0.3 | 26.0±0.4 |
| | GLaD (rand G) | **30.1±0.5** | 27.3±0.2 | 28.0±0.9 | 21.2±0.6 | 26.6±0.5 |
| | GLaD (trained G) | 26.0±0.7 | **27.6±0.6** | 28.2±0.6 | 23.4±0.2 | 26.3±0.5 |
| | **LD3M** (trained G) | 27.2±0.8 | 26.6±0.9 | **31.5±0.3** | **29.0±0.2** | **28.6±0.6** |
| DM | pixel space | 22.9±0.2 | 22.2±0.7 | 23.8±0.5 | 21.3±0.5 | 22.6±0.5 |
| | GLaD (rand G) | 23.7±0.3 | 21.7±1.0 | 24.3±0.8 | 21.4±0.2 | 22.8±0.6 |
| | GLaD (trained G) | 25.1±0.5 | **22.5±0.7** | 24.8±0.8 | 23.0±0.1 | **23.8±0.5** |
| | **LD3M** (trained G) | **27.2±0.4** | 17.0±0.7 | **25.4±0.4** | **23.8±0.3** | 23.4±0.5 |

Table 4: CIFAR-10 Performance on unseen architectures, IPC=1.

**Cross-Architecture Evaluation and different Pixel Spaces, 256x256.** As shown in Table 3, a similar pattern of improved performance is evident. LD3M ($T = 10$) achieved all top-3 results per subset. To explore the versatility of the LDMs that can be used in LD3M, we experimented with various training configurations, including models pre-trained on ImageNet, FFHQ, and a randomly initialized model (no pre-training). These varied LDM configurations demonstrate improved cross-architectural generalization compared to GLaD and are notably better than direct pixel-space distillation.

**CIFAR-10.** Like Cazenavette et al. (2023), we evaluated LD3M on CIFAR-10 in Table 4. While it presents a less challenging task, we observe that LD3M is still on par with DM while significantly better with DC.

**Initializations.** In Table 5, we investigate the influence of different initializations, either Gaussian noise or initialized with a randomly selected image. In conclusion, we can see that LD3M's performance significantly benefits from real images and the straightforward initialization process, especially for the distillation algorithms DM and DC, while MTT seems more robust.

**Visual Comparison, 128x128.** As presented in Figure 3, visual comparisons depict the results of GLaD and the LD3M. These comparisons reveal that the generated images by LD3M are notably more abstract and exhibit higher contrast than those produced by GLaD. This highlights a more flexible learning space for dataset distillation algorithms.

**Visual Comparison, 256x256.** Figure 4 shows $256 \times 256$ resolution results under different initializations. Again, LD3M generates visually distinctive synthetic images with sharper contrast and richer abstractions, which, despite their non-realistic appearance, encode essential class-specific information that improves generalization across unseen architectures. Surprisingly, LD3M generates more stable images under different initializations, as GLaD's generated images vary more based on the dataset used for pre-training the generator. GLaD's generation becomes more noisy towards random initialization (see from left to right). Additional visualizations are provided in the appendix.

| Dist. Method | Dist. Space | AlexNet | ResNet18 | VGG11 | ViT | Average |
|---|---|---|---|---|---|---|
| MTT | Gauss. noise | 28.7±1.6 | 34.1±1.5 | 32.2±0.6 | 29.1±1.8 | 31.0±1.4 |
|  | random image | **30.1±1.4** | **35.6±1.4** | **32.2±0.3** | **30.0±1.2** | **32.0±1.3** |
| DC | Gauss. noise | 13.1±1.5 | 11.6±1.8 | 13.8±2.2 | 13.7±2.7 | 13.1±2.1 |
|  | random image | **31.6±1.3** | **30.4±0.6** | **31.8±1.2** | **37.7±1.5** | **32.9±2.1** |
| DM | Gauss. noise | 13.4±2.0 | 12.4±1.8 | 13.4±1.4 | 14.4±2.1 | 13.4±1.8 |
|  | random image | **31.9±1.3** | **23.2±2.2** | **25.9±2.0** | **26.1±1.4** | **26.8±1.7** |

Table 5: Performance with different initializations of the latent and conditioning codes per class on ImageNette for the IPC=1 setting with 5,000 iterations. MTT with random initialization reaches comparable performance (still under initialization with randomly selected images and their labels), whereas random initialization leads to synthetic datasets with significantly reduced quality for DC and DM.

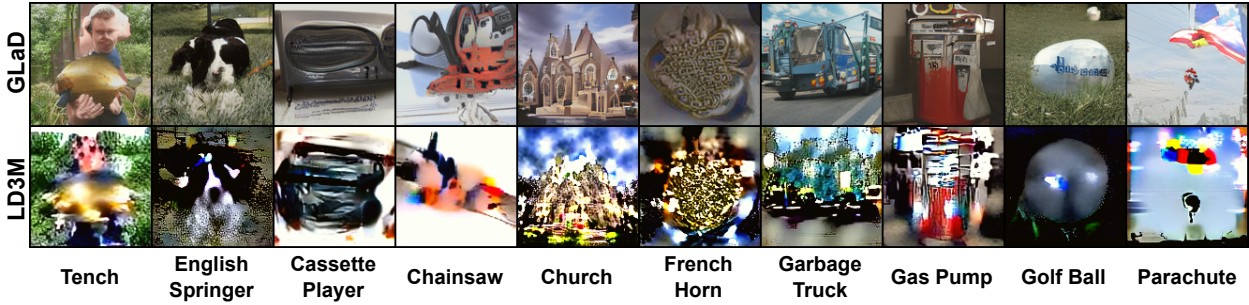

Figure 3: Visual comparison of LD3M versus GLaD for 1000 iterations with MTT on ImageNette. LD3M produces more abstract and contrast-richer synthetic images.

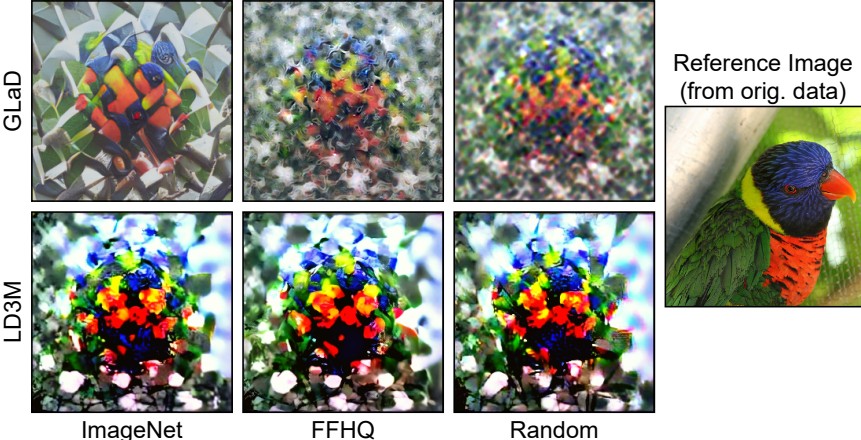

Figure 4: Example $256 \times 256$ images of a distilled class (ImageNet-B: Lorikeet) with differently initialized generators GLaD and LD3M. The various initializations, *i.e.*, which dataset was used for training the generators, are denoted at the bottom.

**Diffusion Steps Analysis.** Our investigation on the trade-off between runtime and accuracy is shown in Figure 5. It illustrates the accuracy of LD3M across different maximum time steps $T$ in the diffusion process, specifically evaluated on the ImageNet A-E datasets using the MTT algorithm. Setting the maximum time step up to 40 for the diffusion process results in a notable increase in accuracy. However, performance declines significantly as we progress beyond 40. This decline could be attributed to vanishing gradients during the

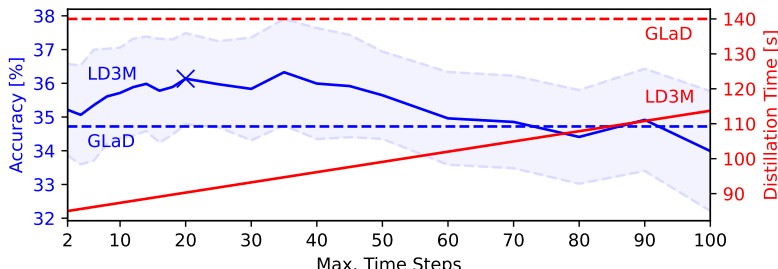

Figure 5: Distilled dataset performance (average of ImageNet A-E via MTT and IPC=1) for different max. time steps employed in the diffusion process with mean and standard deviation (light blue dotted). It illustrates the trade-off between runtime and accuracy over increasing time steps. LD3M shows improved performance for the first 40 steps, while performance deteriorates for greater steps. GLaD needs around 140 seconds per iteration on the same hardware (NVIDIA RTX3090) with 34.72 % average accuracy, shown as dotted lines. Best trade-off marked with "X".

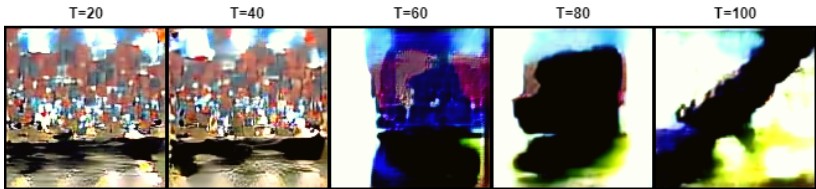

Figure 6: Example images of a distilled class (Freight Car) for different time steps settings. We observe that distillation can collapse for higher time step settings (see the three images on the right).

| Method | All | ImNet-A | ImNet-B | ImNet-C | ImNet-D | ImNet-E |
|---|---|---|---|---|---|---|
| GLaD | 35.4±1.3 | 41.8±1.7 | 42.1±1.2 | 35.8±1.4 | 28.0±0.8 | 29.3±1.3 |
| LD3M (w/o diffusion) | 35.3±1.3 | 40.6±1.9 | 41.9±1.1 | 35.3±1.0 | 29.4±1.4 | 29.5±1.3 |
| LD3M (w/ diffusion) | 36.5±1.3 | 42.3±1.3 | 42.0±1.1 | 37.1±1.8 | 29.7±1.3 | 31.4±1.1 |
| | +1.2+0.0 | +1.7-0.6 | +0.1+0.0 | +1.8+0.8 | +0.3-0.1 | +1.9-0.2 |

Table 6: Performance analysis of LD3M with and without diffusion using the DC distillation algorithm (IPC=1). Without diffusion, LD3M delivers mixed results compared to GLaD, underperforming on datasets like ImageNet-A and ImageNet-C, while achieving comparable overall performance. Incorporating diffusion enables LD3M to consistently outperform GLaD.

diffusion process, suggesting that strategies other than the linear addition of the initial state could avoid the performance decline. An example is shown in Figure 6 for the freight car class from ImageNet-C.

**Hardware Resources and Time.** Overall, LD3M with $T = 20$ needs 574 minutes versus 693 minutes for GLaD. Furthermore, on an A100-40GB, LD3M with $T = 20$ needs 73.58% max. GPU allocation / 29.4GB vs GLaD with 77.94% max. GPU allocation / 31.2GB. Also, LD3M can distill with less memory by adjusting the number of diffusion steps (*i.e.*, from $T = 2$ to $T = 35$ ), which is not doable with GLaD due to the fixed GAN. The ability to tailor the diffusion process according to specific accuracy and runtime opens up new possibilities for dataset distillation across various computational settings: It allows for strategically allocating computational resources, ensuring the models are accurate and practically viable for real-world applications. While dataset distillation remains computationally expensive, LD3M introduces a flexible and significantly lighter alternative to GLaD. Concerning the remaining experiments, we chose $T = 10$ as it represents a good trade-off between accuracy and inference time.

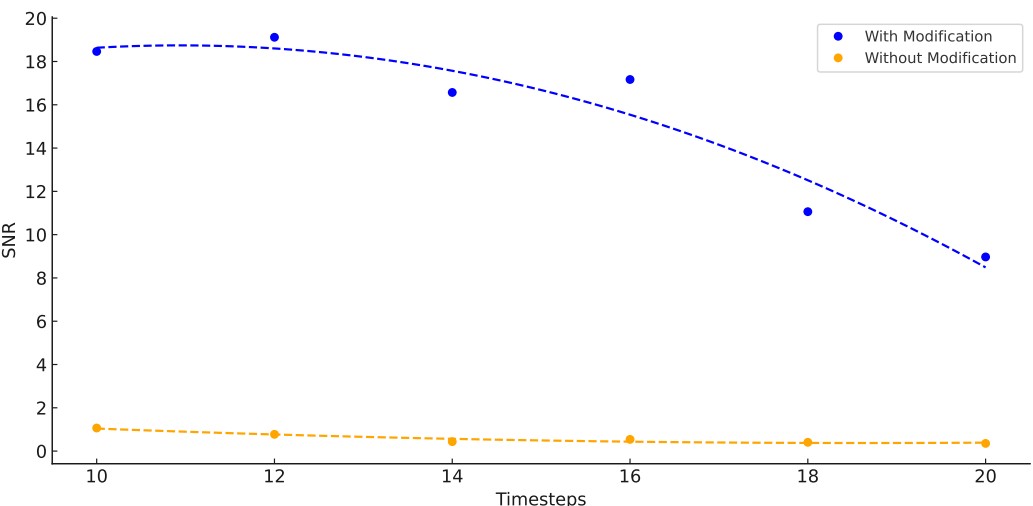

Figure 7: Gradient flow analysis comparing the Signal-to-Noise Ratio (SNR) of gradient norms for LD3M with and without our modification. Diffusion demonstrates a more stable gradient flow, indicating enhanced optimization dynamics. Dashed lines show a polyfit plot to highlight the trends.

**Impact of Diffusion.** Table 6 shows the performance of LD3M with and without diffusion, compared to the baseline GLaD. Without diffusion, LD3M exhibits comparable overall performance to GLaD, achieving a score of 35.3 versus 35.4 but underperforming on specific subsets such as ImageNet-A (-1.2), ImageNet-B (-0.1) and ImageNet-C (-0.5). With the inclusion of diffusion, LD3M demonstrates a clear improvement, achieving an overall score of 36.5 (+1.1 over GLaD and +1.2 over LD3M without diffusion).

**Improved Gradient Flow.** To investigate the impact of the modified Equation 12, we measured the gradient norms of the latent code during distillation for both cases, with and without the incorporation of the initial state $\mathbf{z}_T$. The increase from the non-modified version to the modified version is approximately 66.87 %. The results visualized in Figure 7 show that incorporating our modified diffusion process stabilizes the gradient flow, as indicated by the higher Signal-to-Noise Ratio (SNR) of gradient norms across timesteps, with a trend of descending gradient flow for higher timesteps. This outcome emphasizes the importance of the modified formula for combating vanishing gradients.

**Component Analysis.** In our initial experiments, we evaluated the impact of incorporating different aspects of the LDM, as described in our methodology section and shown in Table 7. We used LD3M on ImageNette and MTT for 1,000 distillation steps and an Image Per Class (IPC) of 1. We also compared against GLaD and evaluated the results with unseen architectures. By applying only learnable conditioning, the resulting performance of LD3M is just roughly five p.p. above random chance. However, the real breakthrough was observed when we allowed the latent representation within the LD3M to be learnable. This adjustment significantly improved accuracy across all models but still lagged behind those achieved with GLaD. Another enhancement was implementing the modified formula for calculating the intermediate states $\mathbf{z}_t$ in the LDM. This involved incorporating the initial state $\mathbf{z}_T$ into the calculation of intermediate states $\mathbf{z}_t$, enabling LD3M to surpass the GLaD performance.

## 5 Related Work

**Diffusion Models vs GANs.** The study by Dhariwal & Nichol (2021) showed that diffusion models outperform GANs in image generation tasks. Since then, diffusion models have impacted fields like image restoration (Moser et al., 2023a;b; Saharia et al., 2022; Li et al., 2022), layout-to-image (Zheng et al., 2023), inpainting (Lugmayr et al., 2022; Yu et al., 2023), medical imaging (Müller-Franzes et al., 2022; Namatevs et al., 2023) and more (Yang et al., 2023; Moser et al., 2024; Bar-Tal et al., 2023; Mukhopadhyay et al., 2023).

| Method | All | AlexNet | VGG-11 | ResNet-18 | ViT |
|---|---|---|---|---|---|
| GLaD | 26.6±1.6 | 28.7±0.3 | **29.2±1.2** | **30.8±2.9** | 17.8±1.5 |
| LDM learnable conditioning (**c**) | 15.8±1.5 | 14.2±2.6 | 15.1 ±1.6 | 16.5±4.9 | 16.8±4.0 |
| + learnable latent code ($\mathcal{Z}$) | 22.3±2.0 | 22.8±2.0 | 26.3±0.9 | 23.4±3.2 | 17.5±2.0 |
| + enhanced gradient flow (Eq. 12) | **28.1±3.3** | **29.2±1.9** | **29.2±1.2** | 30.6±1.3 | **25.1±1.7** |

Table 7: Different LDM variations with MTT on ImageNette and IPC=1. Tested was the distillation by making only the conditioning learnable (**c**), one with also learnable latent representation ($\mathcal{Z}$), and lastly, one which incorporates the initial latent representation in the calculation of intermediate latent states (Eq. 12).

This development of the superior performance of diffusion models over GANs across various fields inspired this work. Unlike GANs, diffusion models do not require extensive regularization and optimization strategies to mitigate issues like optimization instability and mode collapse (Frolov et al., 2021).

**LDMs for GLaD.** The closest work to ours is Duan et al. (2023), which also uses the pre-trained autoencoder of LDM in GLaD and utilizes straightforward initialization. However, they do not utilize any diffusion, limiting their decoding to that of a basic autoencoder. In addition, their experimental setup diverges from GLaD, complicating direct comparison. For instance, they evaluate latent codes per class (LPC), defined by how many latent vectors would have the same memory as one IPC, *i.e.*, 12 LPC for IPC=1 or 120 LPC for IPC=10. Compared to GLaD and us, we have one latent code for one IPC.

**Diffusion Models in Dataset Distillation.** Another notable mention is the work of Gu et al. (2023), which proposed a Minimax Diffusion process for dataset distillation. While they present a promising hierarchical diffusion control mechanism to enable dataset distillation, it is another dataset distillation algorithm and, therefore, orthogonal to our work. Also, we enable a more straightforward mechanism by linearly adding the initial latent representation to enable distillation. For future work, we see great potential in using our method before applying the Minimax Diffusion process, as LD3M can be applied with any dataset distillation method.

# 6 Conclusion and Future Work

We introduced LD3M, a novel diffusion-based distillation approach. A core innovation is its modified diffusion process to address the challenge of the vanishing gradients arising from the iterative time steps. By enhancing gradient flow, LD3M ensures stable optimization, leading to diverse and expressive synthetic datasets without the computational overhead of GAN-based approaches. The abstract yet class-specific synthetic images generated by LD3M also facilitate better generalization across architectures, significantly improving other traditional methods. Our results show that LD3M surpasses the state-of-the-art approach GLaD and provides enhanced data quality and faster distillation. LD3M also offers more control by adjusting the diffusion steps to balance runtime and quality. Also, the initialization of latent codes is straightforward with an autoencoder, avoiding the need for GAN inversion. In conclusion, LD3M constitutes a transformative step in dataset distillation, broadening its applicability and enhancing its practicality.

For future work, other diffusion models in LD3M and alternative formulations of our modified equation, *i.e.*, non-linear, should be investigated. Another interesting avenue is using sampling mechanisms that reduce the overall amount of time steps needed, such as DPM-Solver Lu et al. (2022), which not only helps reduce the vanishing gradient problem but also can be combined with LD3M.

## Acknowledgment

Will be updated...

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

# A  Appendix

## A.1  Note on the Impact of Equation 12 during the Reverse Process

Our experiments show that the error introduced by these modifications can be minimized through optimization of the latent codes $\mathcal{Z}$ and conditioning codes $\mathbf{c}$. However, we also can indicate this theoretically. The error at each time step $t$ caused by the modifications can be defined as $\epsilon_t = \mathbf{z}_t^{\mathrm{orig}} - \mathbf{z}_t^{\mathrm{mod}}$. Given that the LDM is well-trained, we can approximate $\epsilon_t$ using a first-order Taylor expansion:

$$\epsilon_t \approx J_t(\epsilon_{t-1}) + \eta_t(\epsilon_T)$$

where $J_t$ is the Jacobian matrix of the transformation from $\mathbf{z}_{t-1}$ to $\mathbf{z}_t$, and $\eta_t$ represents the influence of the residual connection from $\mathbf{z}_T$.

Therefore, we approximate the modification loss as:

$$\mathcal{L}_{\mathrm{mod}}(\epsilon_t) = \sum_{t=1}^{T} \mathbb{E}\left[\|J_t(\epsilon_{t-1}) + \eta_t(\epsilon_T)\|^2\right]$$

Since $J_t$ and $\eta_t$ are small due to the well-trained nature of the original LDM, the optimization process reduces $\epsilon_{t-1}$ at each step, leading to a diminishing cumulative error $\epsilon_t$.

The error at each step $t$ is bounded by:

$$\|\epsilon_t\| \leq \|J_t\| \cdot \|\epsilon_{t-1}\| + \|\eta_t\| \cdot \|\epsilon_T\|$$

Given that $\|J_t\|$ and $\|\eta_t\|$ are small, $\|\epsilon_t\|$ diminishes as $t$ decreases. After optimization, the cumulative effect of $\epsilon_t$ on $\mathcal{L}_{\mathrm{mod}}$ is negligible, meaning that the modifications do not significantly impact the quality of the generated images.

This, however, would greatly influence the quality and especially the content of the generated images if not applied in the context of dataset distillation but in a classical image generation setting. Since the reverse process is altered at each step, we observe a tendency for the generated images to become increasingly abstract and unnatural, deviating significantly from the structure and content of the original training dataset.

For example, when we apply the modification without proper tuning on a model trained with the FFHQ dataset, the generated images exhibit exaggerated artifacts and lose the coherence expected in a facial dataset, as shown in Figure 8. As shown in the provided images, the outputs are highly abstract and fail to resemble the characteristics of the FFHQ dataset.

Also, we observed that the abstract images generated by the LDM with our proposed modification exhibit a slight decrease in diversity. Specifically, we measured the average pairwise LPIPS distance of the generated images, obtaining scores of 0.3855 with the modification and 0.4204 without it, indicating a reduction in variability when the modification is applied.

For this reason, our proposed modification is specifically tailored to the context of dataset distillation. Altering the reverse process introduces flexibility in learning abstract representations, but it does so at the expense of image fidelity and content coherence in a conventional image synthesis setting. In the context of dataset distillation, however, this abstraction serves a critical purpose: it enables the generation of synthetic datasets that encapsulate the essential features needed for downstream tasks without overfitting the original data's appearance. This ensures that the distilled datasets generalize well across diverse architectures while maintaining computational efficiency, making our approach uniquely suited for this specific application.

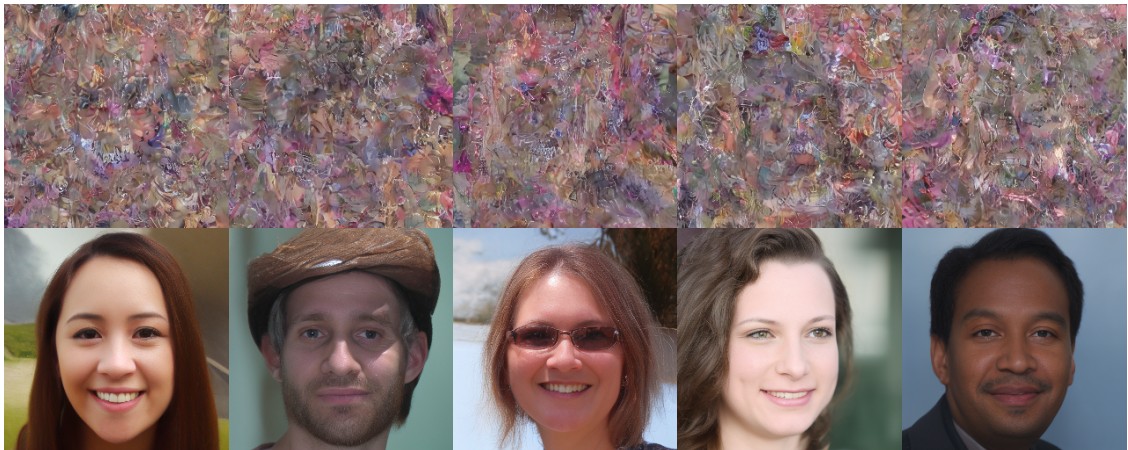

Figure 8: Influence of our modified Equation 12 in a classical image generation setting (unconditional FFHQ). It shows that the residual connections alter the generation process significantly, leading to abstract artifacts and the loss of coherence expected in a facial dataset: **(top)** with modification and **(bottom)** without modification.

| Parameter | Value |
|---|---|
| DSA Augmentations | Color / Crop / Cutout / Flip / Scale / Rotate |
| Iteration (Distillation) | 5,000 ($128 \times 128$) / 10,000 ($256 \times 256$) |
| Momentum | 0.5 |
| Batch Real | 256 |
| Batch Train | 256 |
| Batch Test | 128 |

Table 8: Common hyper-parameters for training the distillation algorithms used in this work.

## A.2 Algorithmic Description

---
**Algorithm 1** Latent Dataset Distillation with Diffusion Models (LD3M)

---
**Input:** randomly selected collection $X_s$, pre-trained encoder $\mathcal{E}$, pre-trained decoder $\mathcal{D}$, pre-trained denoiser $\mu_\theta$, noise levels $\sigma_t$.

$\quad \mathcal{Z} = \mathcal{E}\left(X_s\right)$
$\quad \mathbf{z}_T \sim q(\mathbf{z}_T \mid \mathcal{Z})$
$\quad$ **for** $t = T, \ldots, 1$ **do**
$\quad\quad \varepsilon_t \sim \mathcal{N}(\mathbf{0}, \mathbf{I})$
$\quad\quad \mathbf{z}_{t-1} \leftarrow \left(\left(1 - \frac{t}{T}\right) \cdot \mu_\theta(\mathbf{c}, \mathbf{z}_t, \gamma_t) + \frac{t}{T} \cdot \mathbf{z}_T\right) + \sigma_t^2 \varepsilon_t$
$\quad$ **end for**
$\quad X_{\text{syn}} \leftarrow \mathcal{D}\left(\mathbf{z}_0\right)$
$\quad$ **Return:** $X_{\text{syn}}$

---

## A.3 Hyper-Parameters for Distillation Algorithms

**LDM.** For all our LDM experiments, we set the unconditional guidance scale to default value 3. For $128 \times 128$ images, we used max. time steps of 10, and for $256 \times 256$ images, we used 20.

**DC.** We utilize a learning rate of $10^{-3}$ throughout our DC experiments to update the latent code representation and the conditioning information.

**DM.** In every DM experiment conducted, we adopt a learning rate of $10^{-2}$, applying it to updates of the latent code representation alongside the conditioning information.

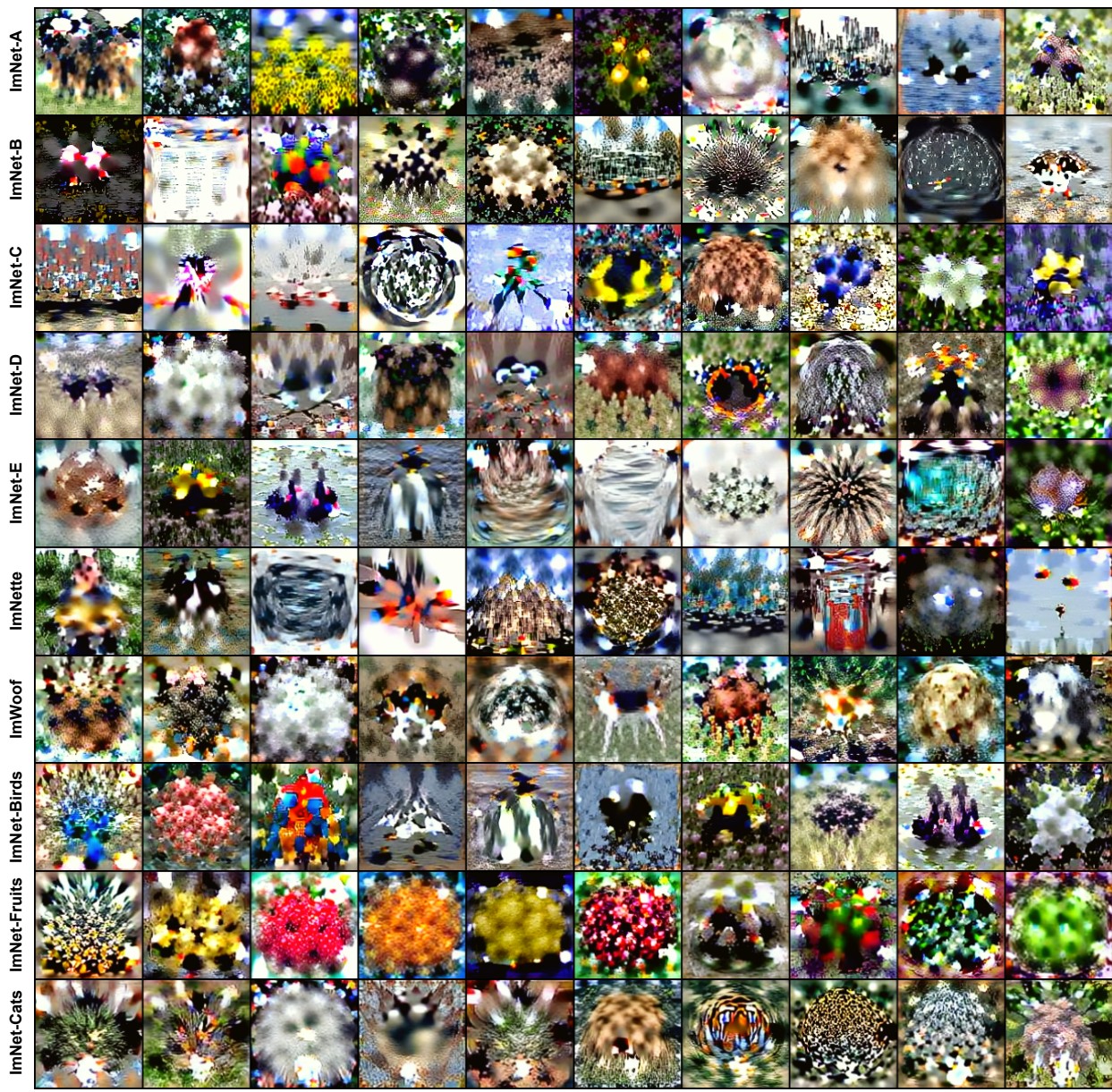

Figure 9: Images distilled by MTT in LD3M for IPC=1.

**MTT.** For MTT experiments, a uniform learning rate of 10 is applied to update the latent code representation and the conditioning information. We buffered 5,000 trajectories for expert training, each with 15 training epochs. We used ConvNet-5 and InstanceNorm. During dataset distillation, we used three expert epochs, max. start epoch of 5 and 20 synthetic steps.

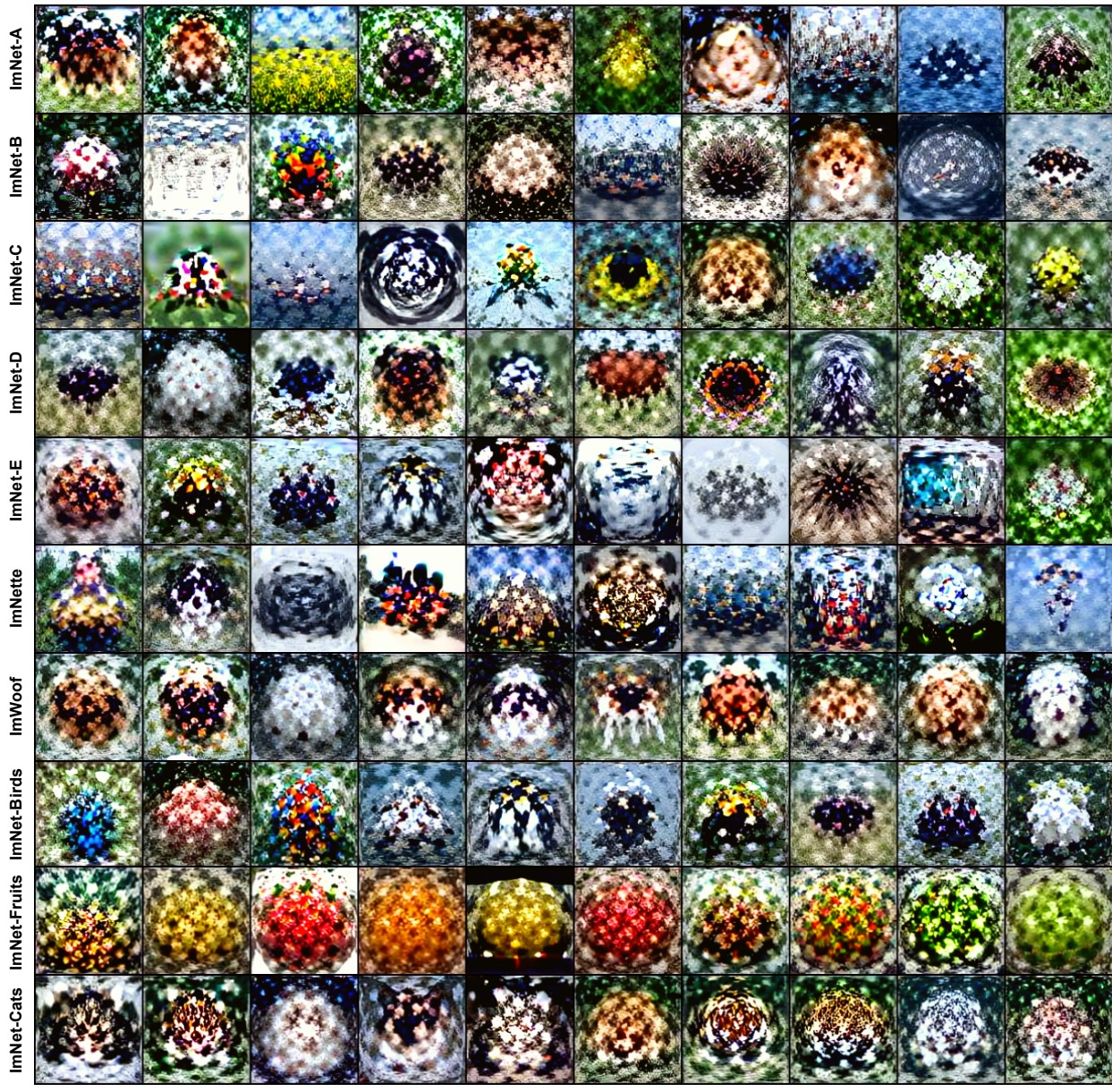

Figure 10: Images distilled by DC in LD3M for IPC=1.

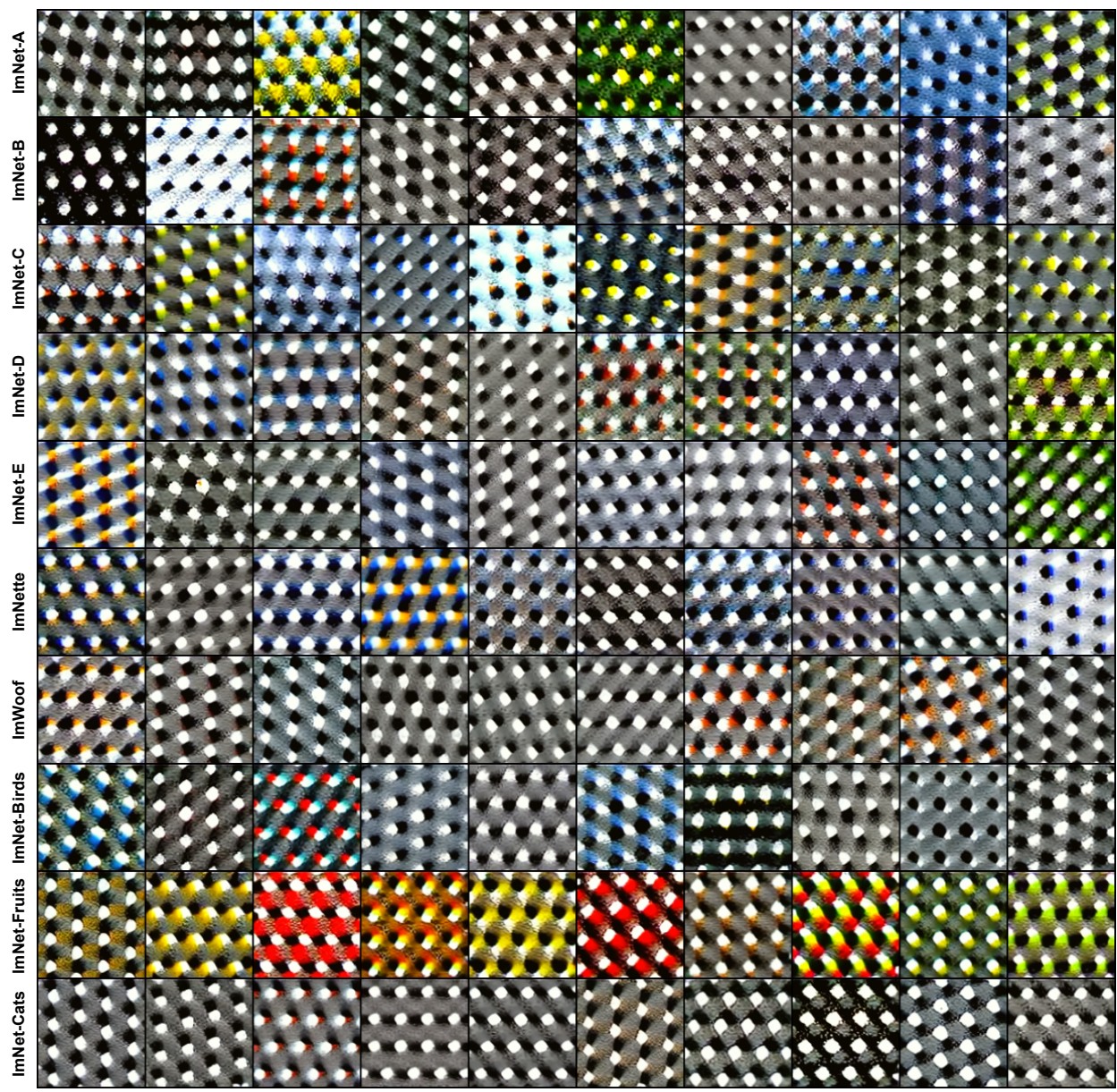

Figure 11: Images distilled by DM in LD3M for IPC=1.

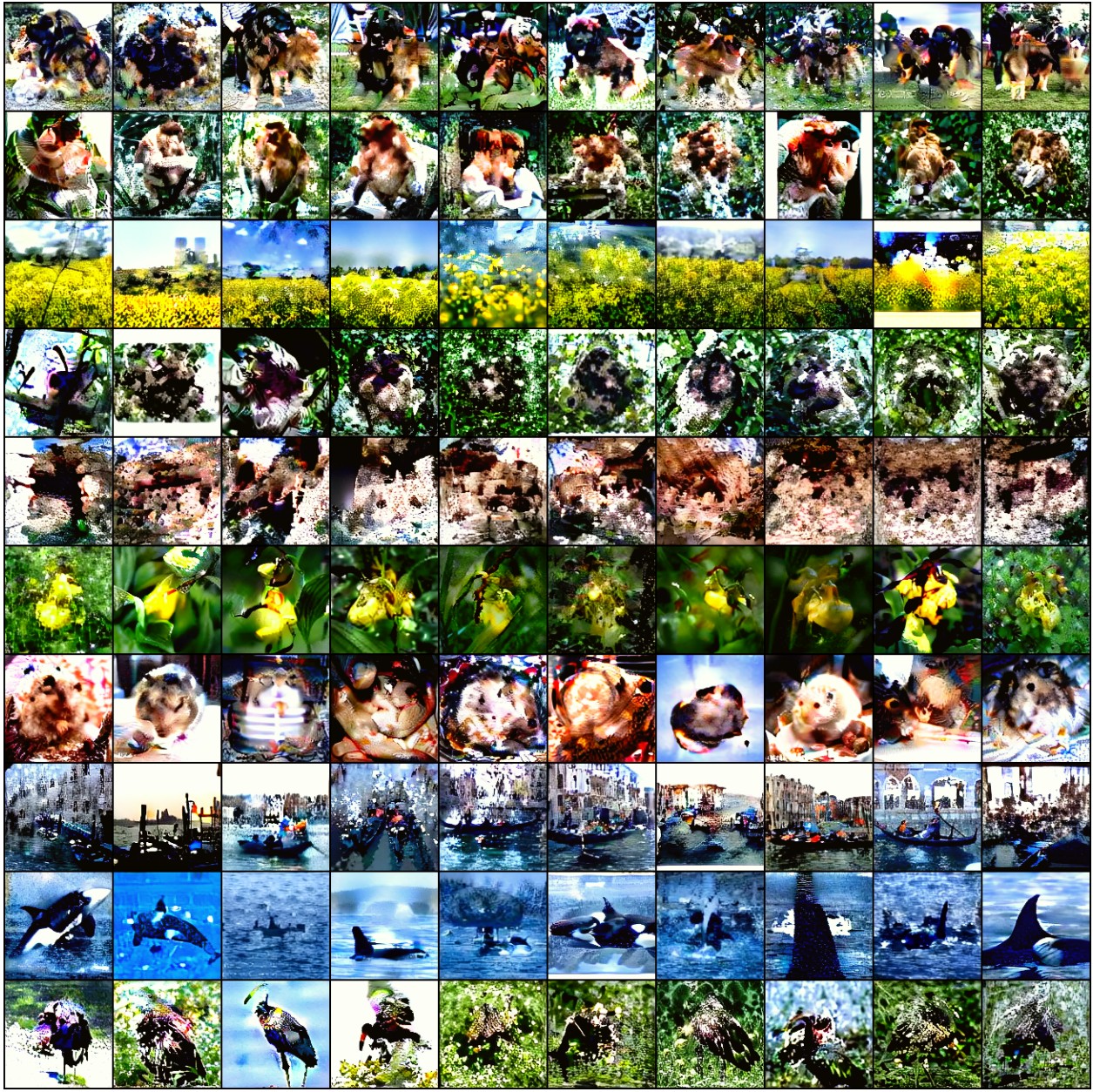

Figure 12: Images distilled by DC in LD3M for IPC=10 and ImageNet-A.

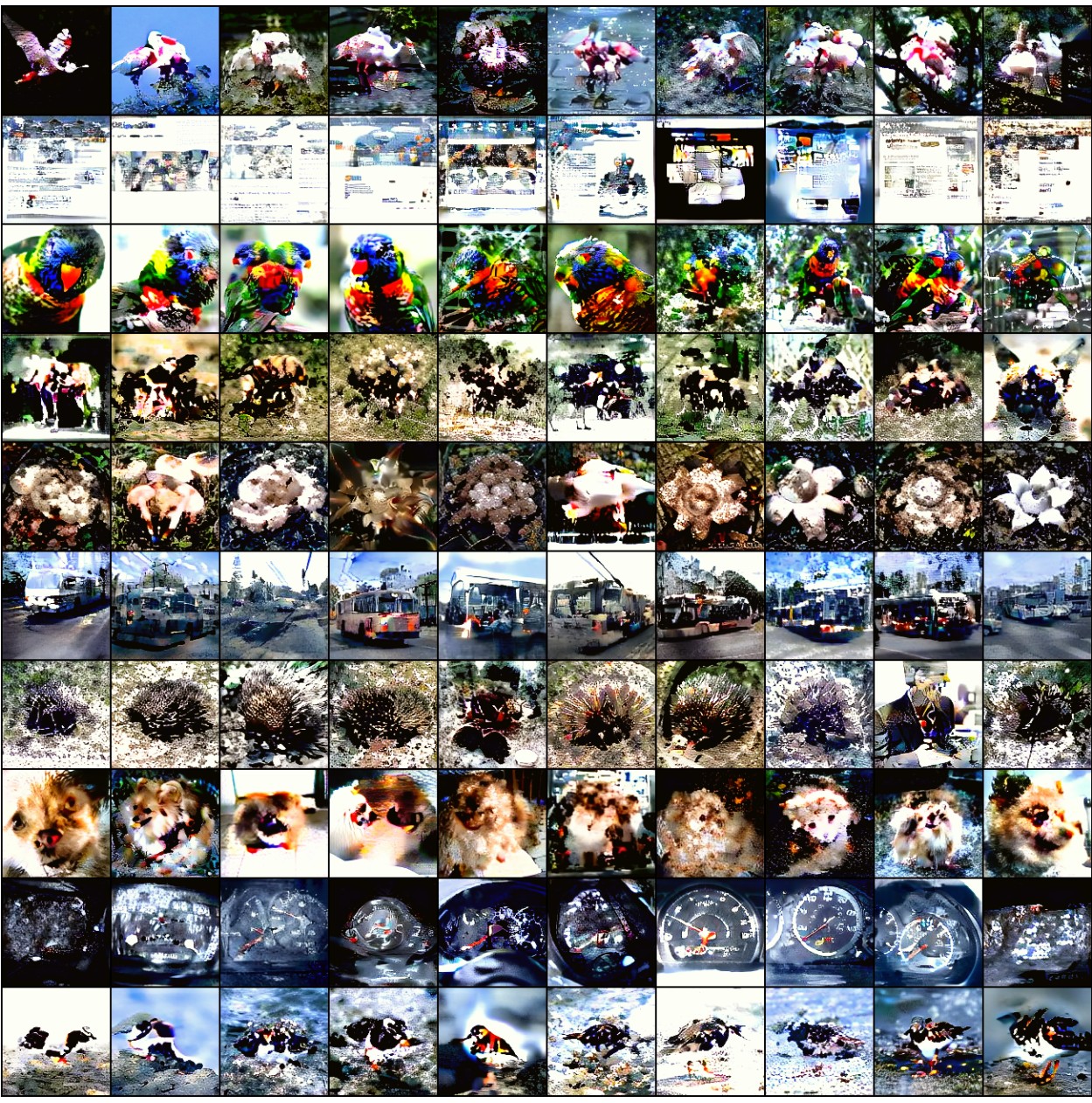

Figure 13: Images distilled by DC in LD3M for IPC=10 and ImageNet-B.

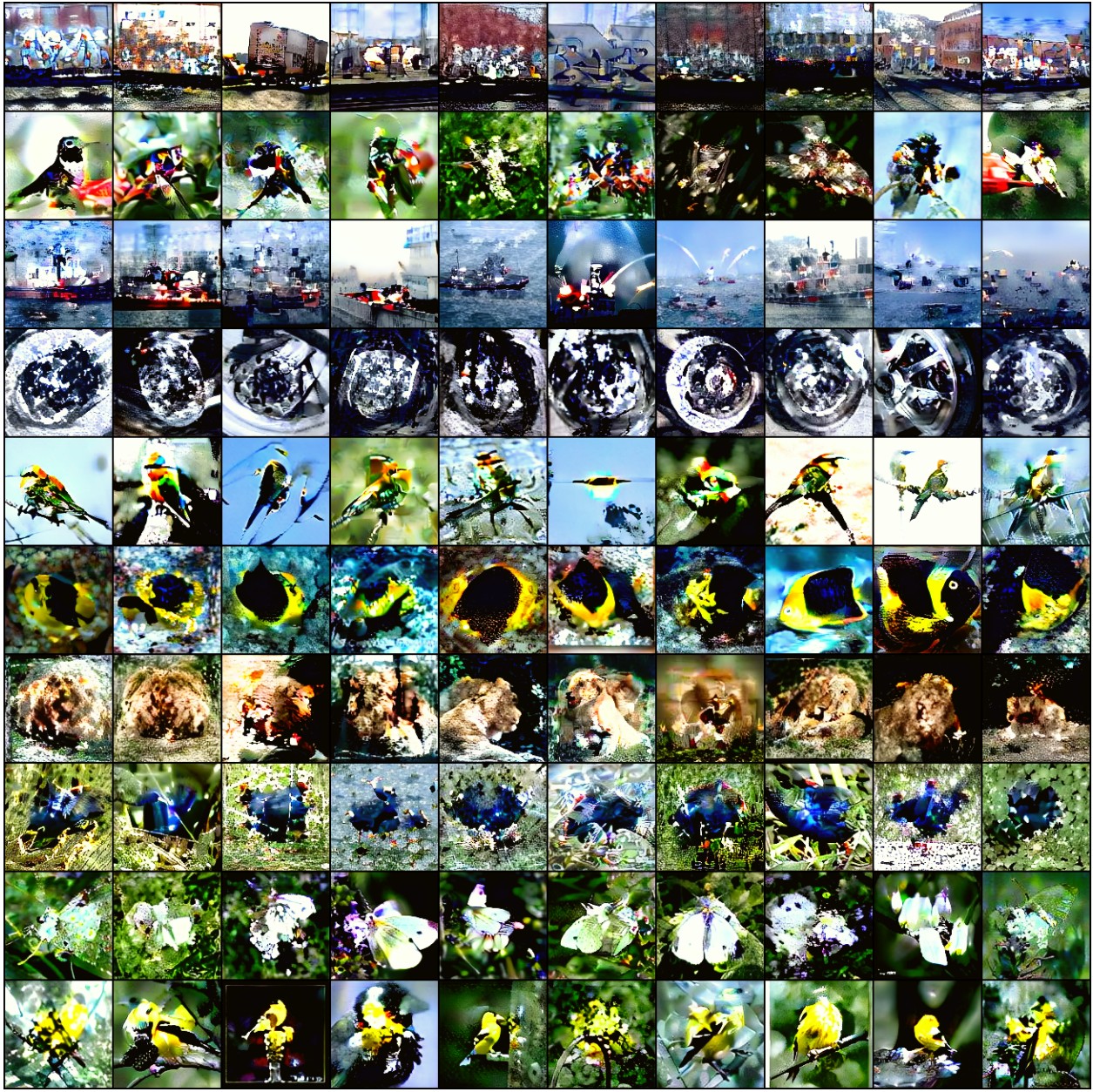

Figure 14: Images distilled by DC in LD3M for IPC=10 and ImageNet-C.

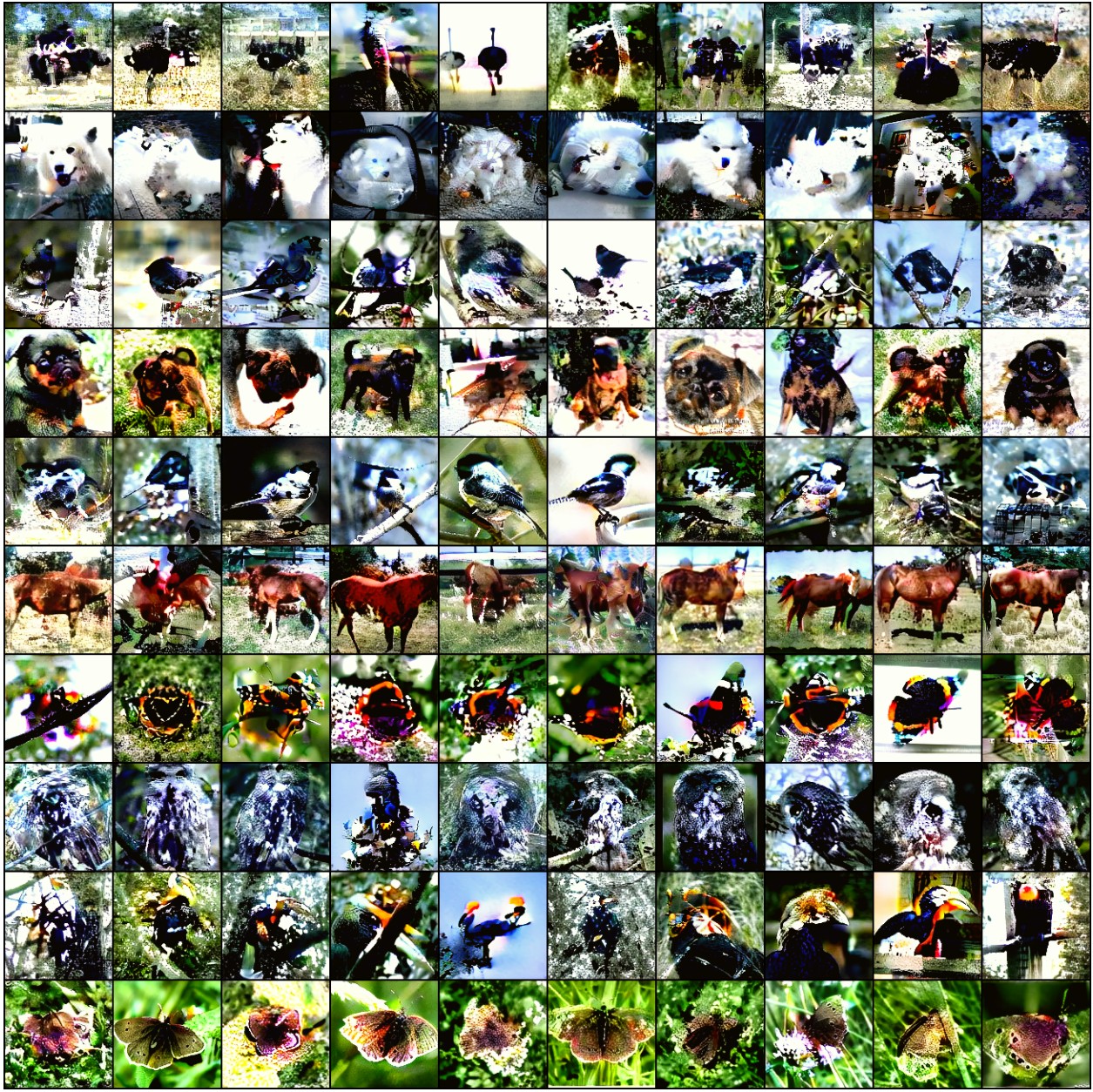

Figure 15: Images distilled by DC in LD3M for IPC=10 and ImageNet-D.

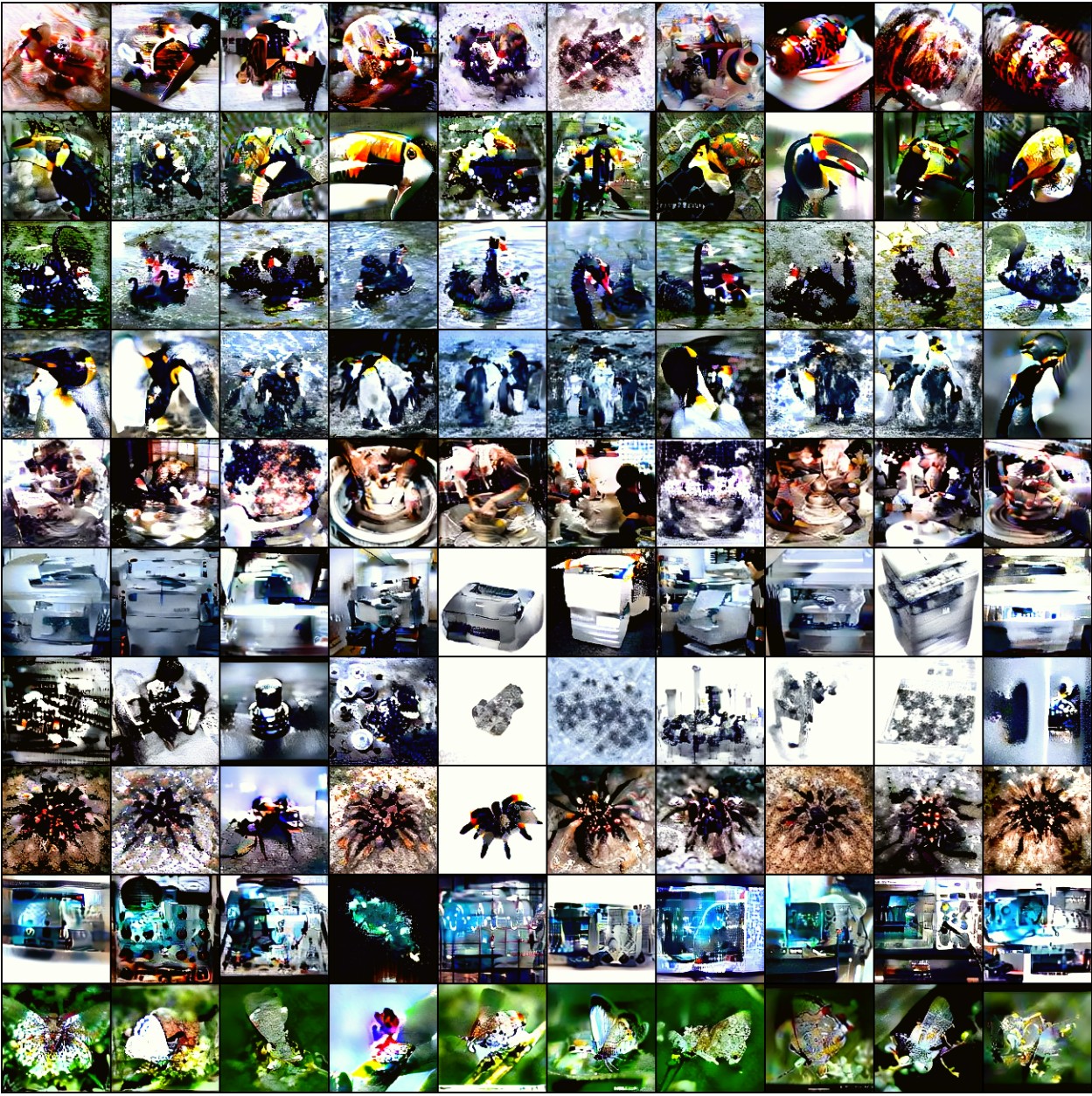

Figure 16: Images distilled by DC in LD3M for IPC=10 and ImageNet-E.

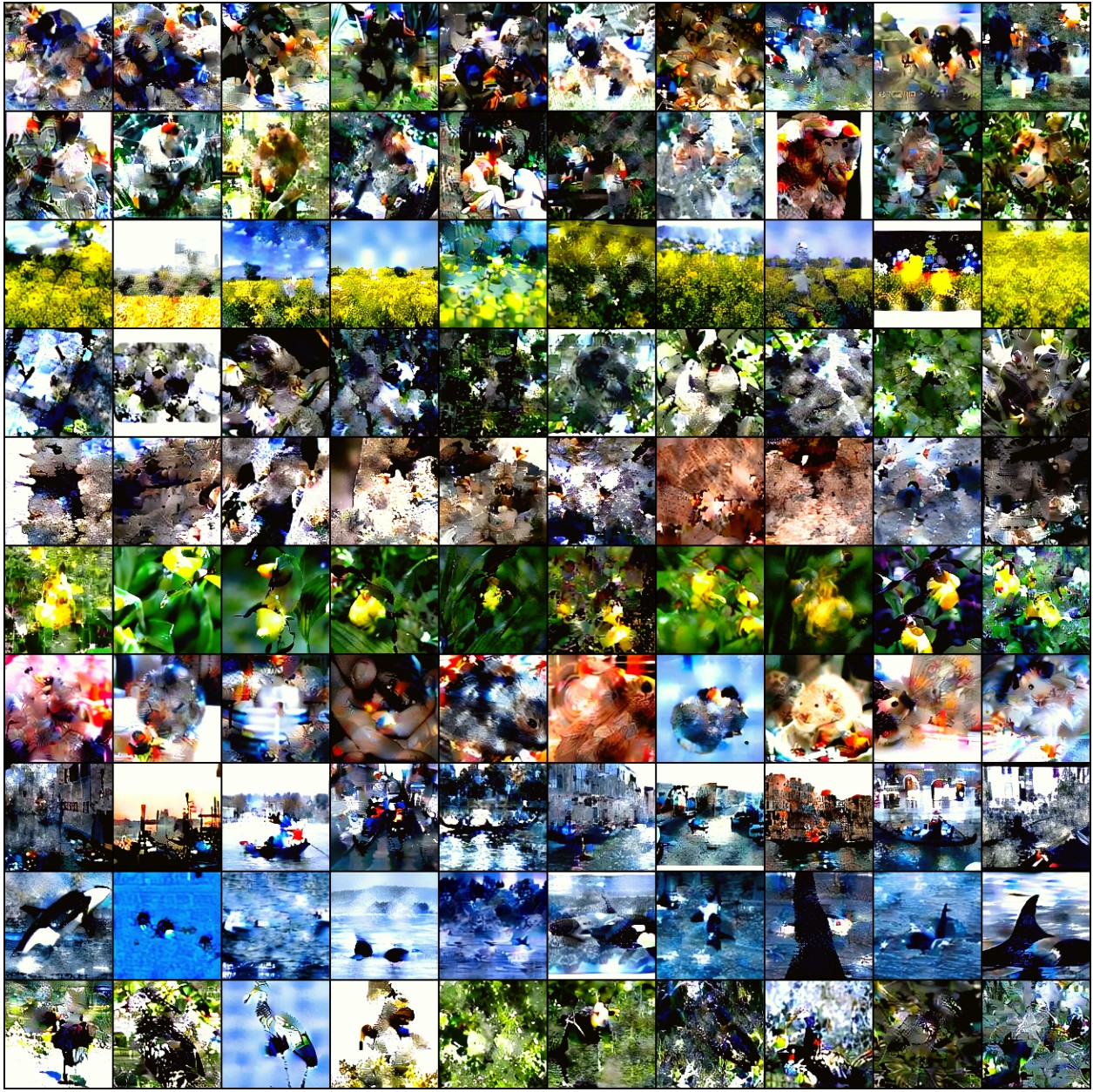

Figure 17: Images distilled by DM in LD3M for IPC=10 and ImageNet-A.

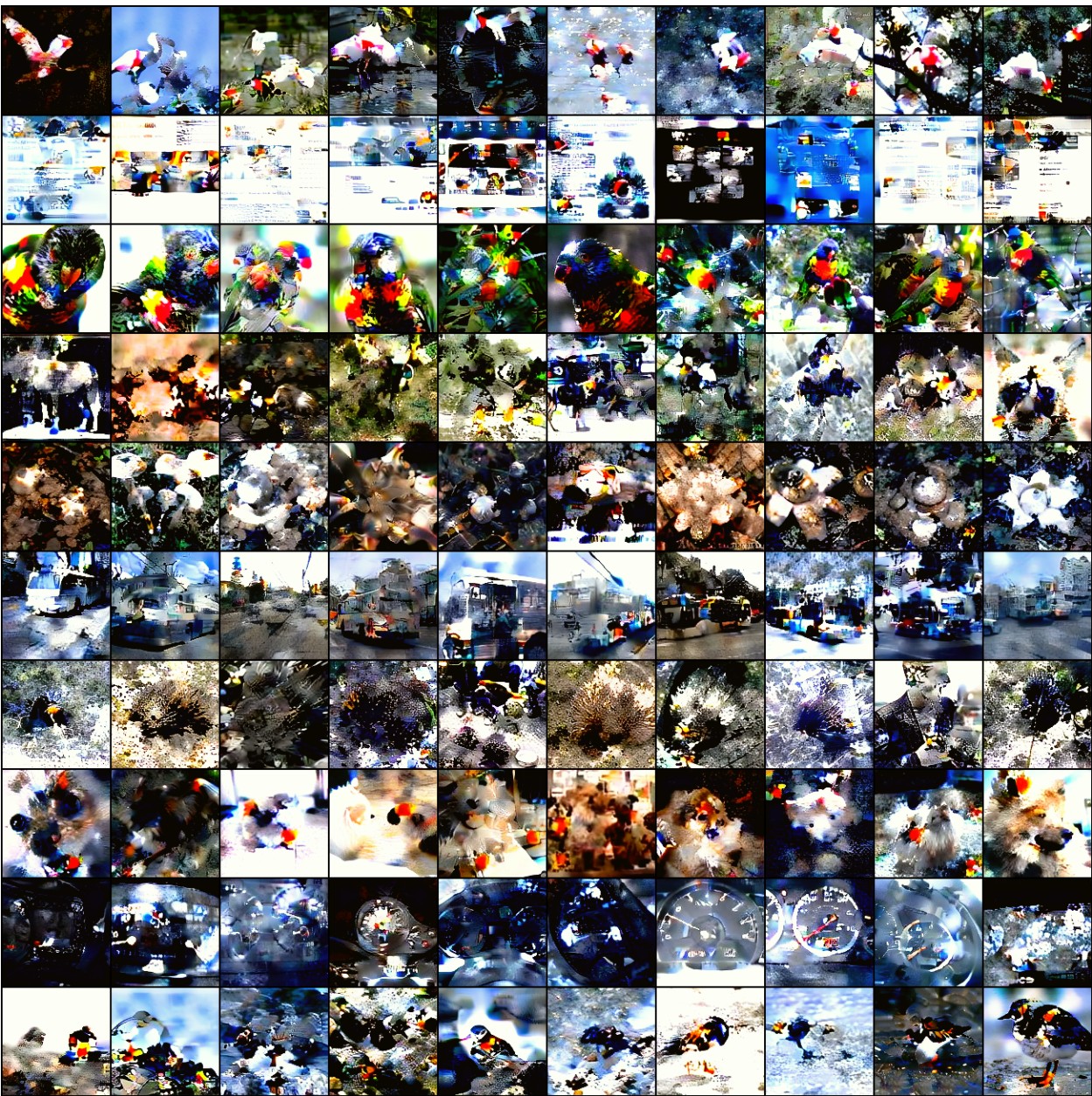

Figure 18: Images distilled by DM in LD3M for IPC=10 and ImageNet-B.

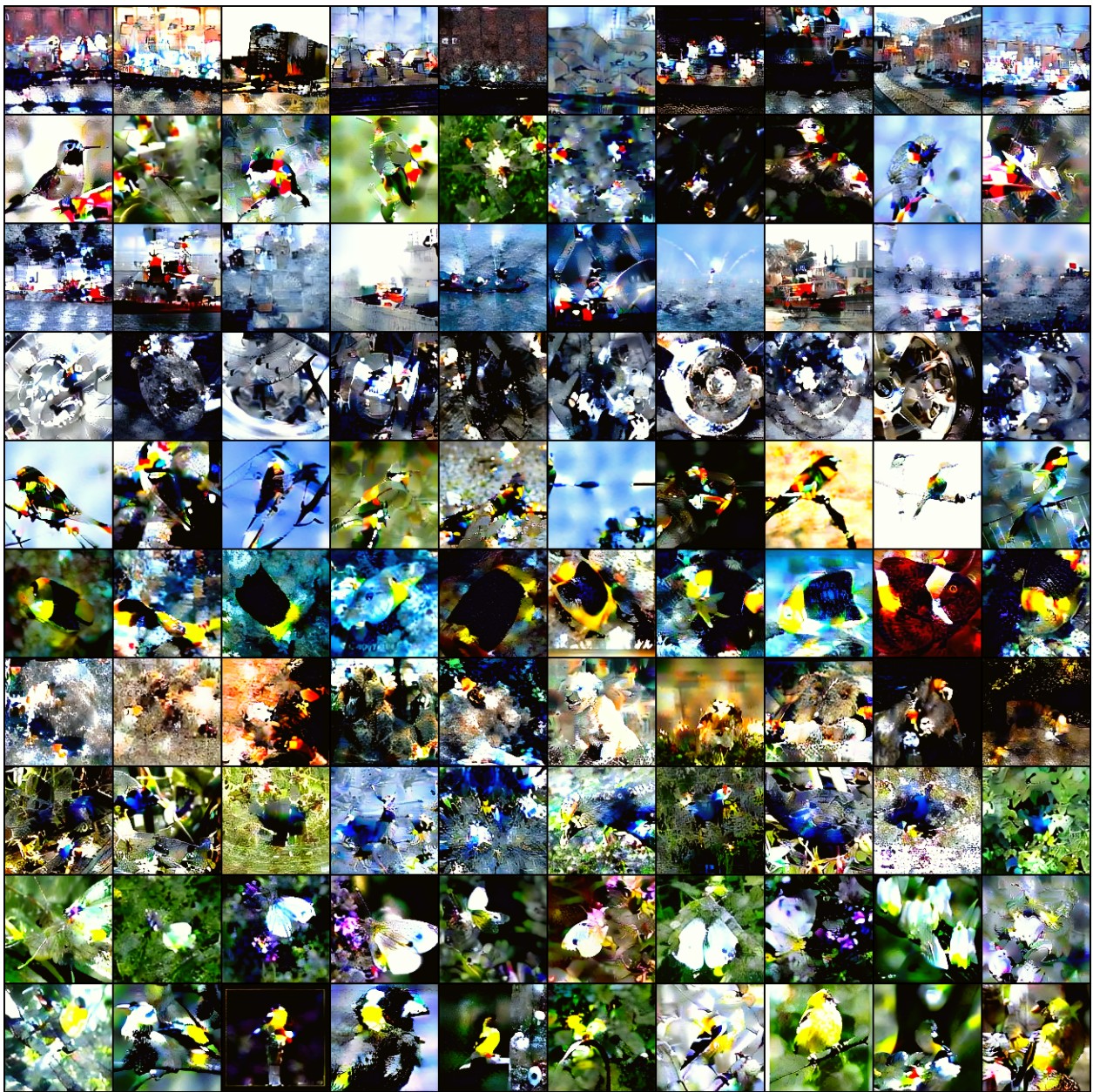

Figure 19: Images distilled by DM in LD3M for IPC=10 and ImageNet-C.

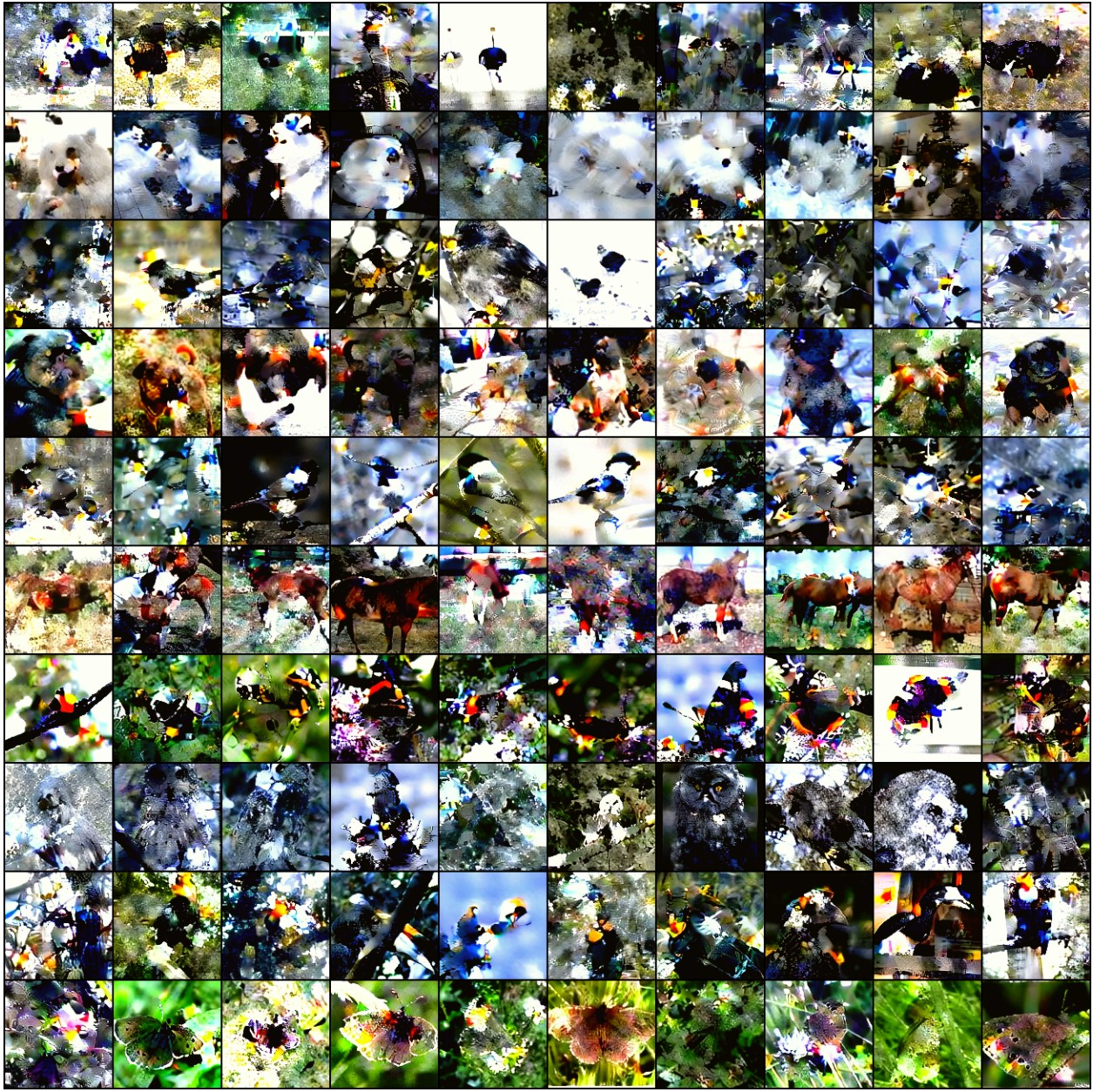

Figure 20: Images distilled by DM in LD3M for IPC=10 and ImageNet-D.

| Dataset | 0 | 1 | 2 | 3 | 4 | 5 | 6 | 7 | 8 | 9 |
|---|---|---|---|---|---|---|---|---|---|---|
| ImageNet-A | Leonberg | Probiscis Monkey | Rapeseed | Three-Toed Sloth | Cliff Dwelling | Yellow Lady's Slipper | Hamster | Gondola | Orca | Limpkin |
| ImageNet-B | Spoonbill | Website | Lorikeet | Hyena | Earthstar | Trollybus | Echidna | Pomeranian | Odometer | Ruddy Turnstone |
| ImageNet-C | Freight Car | Hummingbird | Fireboat | Disk Brake | Bee Eater | Rock Beauty | Lion | European Gallinule | Cabbage Butterfly | Goldfinch |
| ImageNet-D | Ostrich | Samoyed | Snowbird | Brabancon Griffon | Chickadee | Sorrel | Admiral | Great Gray Owl | Hornbill | Ringlet |
| ImageNet-E | Spindle | Toucan | Black Swan | King Penguin | Potter's Wheel | Photocopier | Screw | Tarantula | Sscilloscope | Lycaenid |
| ImageNette | Tench | English Springer | Cassette Player | Chainsaw | Church | French Horn | Garbage Truck | Gas Pump | Golf Ball | Parachute |
| ImageWoof | Australian Terrier | Border Terrier | Samoyed | Beagle | Shih-Tzu | English Foxhound | Rhodesian Ridgeback | Dingo | Golden Retriever | English Sheepdog |
| ImageNet-Birds | Peacock | Flamingo | Macaw | Pelican | King Penguin | Bald Eagle | Toucan | Ostrich | Black Swan | Cockatoo |
| ImageNet-Fruits | Pineapple | Banana | Strawberry | Orange | Lemon | Pomegranate | Fig | Bell Pepper | Cucumber | Granny Smith Apple |
| ImageNet-Cats | Tabby Cat | Bengal Cat | Persian Cat | Siamese Cat | Egyptian Cat | Lion | Tiger | Jaguar | Snow Leopard | Lynx |

Table 9: Class listings for our ImageNet subsets.

