# OpenReview forum: "LD3M: Latent Dataset Distillation with Diffusion Models"
_TMLR — Rejected by TMLR_

### Review · Reviewer_spgB · 2024-11-09

**Summary Of Contributions:**

* Approach: LD3M introduces a unique method that combines a modified diffusion process with latent dataset distillation, improving the generalization of synthetic datasets across unseen architectures.
* High-Resolution Image Handling: Tackles dataset distillation for higher resolutions (128x128, 256x256), a limitation in previous methods.
* Enhanced Gradient Flow: Implements a modified diffusion model with residual connections, ensuring better gradient propagation and more efficient training.
* Flexible Framework: Integrates with various pre-trained diffusion models without requiring additional fine-tuning, demonstrating versatility and extensibility.

**Audience:**

Yes

**Broader Impact Concerns:**

No concerns

**Claims And Evidence:**

Yes

**Requested Changes:**

* Diversity: The authors could explore how the use of skip connections affects diversity in the generated synthetic images. A note on diversity could also be sufficient as the paper is quite extensive in the current form. Additionally, this could be investigated through an experiment measuring the distance between embeddings in a classification model, such as MobileNet or Inception, particularly when the number of images per class (IPC) exceeds 1 on GlaD and L3DM. Or, they might consider evaluating the performance on challenging examples (e.g., high-loss examples) by comparing synthetic datasets generated by GLaD and L3DM or an experiment in similar spirit.
* Algorithm: The authors should consider adding an algorithm block to the main paper or appendix for readers who need further details. One option could be to provide a brief summary in the main paper with a reference to the appendix.

Minor:

* Correct URL Formatting: In the citation for Jeremy Howard. A smaller subset of 10 easily classified classes from imagenet, and a little more french, remove the space in the URL:
    Change: https://github. com/fastai/imagenette
    To: https://github.com/fastai/imagenette

* Duplicate Reference Removal:

    Chitwan Saharia, Jonathan Ho, William Chan, Tim Salimans, David J Fleet, and Mohammad Norouzi. Image super-resolution via iterative refinement. arXiv:2104.07636, 2021 appears twice. Consolidate this reference to avoid duplication.

**Strengths And Weaknesses:**

Strengths:
* Generalization: Generally outperforms other methods in cross-architecture generalization, enhancing the applicability of distilled datasets.
    Efficiency: Pre-trained encoder allows for efficient latent code initialization, reducing computational complexity compared to GAN-based methods.
* Resolution Capabilities: Superior performance on high-resolution datasets, providing broader use cases in dataset distillation.
* Extensibility: Compatible with emerging diffusion models, making it future-proof for evolving generative models.
* Presentation: The paper is well written and easy to follow

Weaknesses:

Skip connections: The authors introduce skip connections in the reverse process. This has two implications
* Reduced Diversity: By reinforcing strong gradient pathways, skip connections might reduce the diversity of latent features learned at different steps, which is crucial in generative tasks to ensure diverse synthetic outputs. This can lead to poor performance on the tail or susceptibility to adversarial attacks.
* Dependency on Initialization: Improper initialization of skip connections can bias the learning process, potentially causing the model to converge to suboptimal solutions.

---

> ### Author Response · Authors · 2024-11-21
>
> We thank the reviewer for taking the time to review our paper carefully and for the valuable comments. Please find our responses below:
>
> - **Diversity**: We have conducted an additional analysis to evaluate how the use of skip connections in our modified LDM affects diversity (in classical image generation settings). Specifically, we measured the pairwise LPIPS between generated images for both the modified and unmodified LDM. The results indicate that the skip connections slightly reduce diversity, as evidenced by a decrease in the average LPIPS from 0.4284 (unmodified) to 0.3855 (modified). We have included this analysis in the revised version of the paper’s supplementary material and believe it provides valuable insights into the impact of skip connections on the generative process.
> - **Algorithm**: We have included a detailed algorithmic description in the supplemental materials (see A2) and added a reference to it in the methodology section for clarity and accessibility.
> - **Minor**: We have corrected the URL and removed the duplicate reference.

---

> > ### Comment · Reviewer_spgB · 2024-11-26
> >
> > Thank you for your response. I have read the updated paper, my questions were addressed.

---

### Review · Reviewer_9a7u · 2024-11-11

**Summary Of Contributions:**

This paper proposes to incorporate a generative prior into dataset distillation with latent diffusion models. The main methodological contribution is a modified diffusion process with residual connections to address vanishing gradients when backpropagating through diffusion.

**Audience:**

Yes

**Broader Impact Concerns:**

None.

**Claims And Evidence:**

No

**Requested Changes:**

Please see the weaknesses section.
In addition, I feel it would be helpful to add a discussion on how hyperparameter T was set depending on the image resolution.

**Strengths And Weaknesses:**

**Strengths:**
* The main strength of the paper is that it is shown to be compatible with 3 distinct distillation frameworks and is also effective.
* The experiments follow standard settings and are comprehensive including evaluation across unseen architectures.
* The visualisations in the paper are interesting and help to understand the distillation process.

**Weaknesses:**

The main weakness, in my opinion, is that the paper does not clearly justify the following claims:
> "... LD3M ... fully leverages the diffusion process. In contrast, others rely solely on pre-trained decoders or averaging encoded codes, limiting their ability to truly learn representations."

Can you please provide evidence showing the benefit of leveraging the diffusion process? For example, if you just optimize the latent-code (as in Duan et al (2023)) without using the diffusion process, what would the performance be like? Further, the diffusion process is altered due to residual connections and it is not intuitively clear what diffusion process is really being used as the generative prior?

> "By enhancing gradient flow through residual connections, LD3M not only accelerates the distillation process but
also sets a new foundation for future innovations in generative modeling."

I do not see any evidence of improved image quality due to the residual connections. Can you please substantiate this evidence with an evaluation of generative-modeling with residual connections (e.g., FID/Inception-score)? To my understanding, this change alters the reverse-diffusion process and I believe that the probability distribution of this altered reverse-diffusion is different from that of the original data-distribution.

> "A significant challenge in making the latent code Z, which leads to the initial state zT , learnable arises from the numerous time steps during the backward diffusion process, which can take T = 200 steps or more."

While I understand the intention of this statement, the latent-distillation algorithm does not use values of T other than 10 or 20. In the ablation study (Fig. 6), we observe that the residual connections does not help beyond T=40. Could you please clarify this?

> One of the most significant issues is the vanishing gradient problem that can occur during distillation due to the many iterative time steps in the diffusion process.

It would be helpful to demonstrate the vanishing gradient quantitatively. Also, do you use stochastic reverse-diffusion solvers (e.g., DDPM)? In that case, I suspect that some of the optimization difficulties should be attributed to stochasticity. Furthermore, recent techniques (e.g., DPM-Solver) can achieve impressive image quality in very few steps and that appears to be a better solution (both, theoretically and practically) to address vanishing gradients than residual connections.

---

> ### Author Response · Authors · 2024-11-21
>
> We sincerely thank you for the detailed feedback. Please examine the updated submission and find our responses below.
>
> > Can you please provide evidence showing the benefit of leveraging the diffusion process? For example, if you just optimize the latent-code (as in Duan et al (2023)) without using the diffusion process, what would the performance be like?
>
> This is an interesting suggestion to broaden the understanding of LD3M. We added another paragraph to the results section of our paper (Impact of Diffusion) with the suggested experiments (IPC=1 on ImNet A-E). These new experiments show that the diffusion process does indeed improve the distillation process. Without the diffusion process, LD3M has mixed results (sometimes better than GLaD, sometimes worse). With diffusion, we consistently outperform GLaD.
> |Method|All|A|B|C|D|E|
> |-|:-:|-:|-:|-:|-:|-:|
> |GLaD|35.4+-1.3|41.8+-1.7|42.1+-1.2|35.8+-1.4|28.0+-0.8|29.3+-1.3|
> |LD3M (w/o diffusion)|35.3+-1.3|40.6+-1.9|41.9+-1.1|35.3+-1.0|29.4+-1.4|29.5+-1.3|
> |LD3M (w diffusion)|36.5+-1.3|42.3+-1.3|42.0+-1.1|37.1+-1.8|29.7+-1.3|31.4+-1.1|
> ||+1.2+0.0|+1.7-0.6|+0.1+0.0|+1.8+0.8|+0.3-0.1|+1.9-0.2|
>
> > Can you please substantiate this evidence with an evaluation of generative-modeling with residual connections (e.g., FID/Inception-score)? To my understanding, this change alters the reverse-diffusion process and I believe that the probability distribution of this altered reverse-diffusion is different from that of the original data-distribution.
>
> In the supplemental materials (A1), we now provide this analysis using unconditional image generation on FFHQ. It is indeed true that our modified formulation of the reverse-diffusion process significantly alters the probability distribution, as evidenced by the generated images being highly abstract and diverging from FFHQ (also in supplemental materials, Figure 8). Thus, the FID/IS is obsolete with respect to classical image generation due to the alteration, which focuses on improved the learning of latent codes instead of reproducing the learned data distribution of FFHQ. We reinforced in the text that our approach is specifically tailored for the context of dataset distillation, where the goal was to enable effective learning through the diffusion process. In addition, we emphasize that our approach is not designed for, nor should it be applied to, classical image generation tasks where fidelity to the original data distribution is critical.
>
> > While I understand the intention of this statement, the latent-distillation algorithm does not use values of T other than 10 or 20. In the ablation study (Fig. 6), we observe that the residual connections does not help beyond T=40. Could you please clarify this?
>
> We have clarified this section by emphasizing that our results demonstrate the benefits up to T=40 and we specifically highlight that our experiments focus on T=10 and T=20.
>
> > It would be helpful to demonstrate the vanishing gradient quantitatively.
>
> We have added a new paragraph to the main paper (Improved Gradient Flow). In this paragraph, we present an analysis of the average Signal-to-Noise Ratio (SNR) of gradient norms (mean/std) for different time steps, comparing results  with and without our modification in a visualization (see Figure 7). This analysis highlights the improved gradient flow achieved with our method and demonstrates the low SNR for LD3M in the absence of our proposed modification (an indication of vanishing gradients). We believe this quantitative addition addresses your concern. In the following table, you can find the SNR values for the corresponding figure:
> |Method|T=10|T=12|T=14|T=16|T=18|T=20|
> |-|:-:|-:|-:|-:|-:|-:|
> |w/o modification|1.07|0.77|0.43|0.54|0.41|0.35|
> |w/ modification|18.46|19.12|16.57|17.17|11.06|28.97|
>
> > Also, do you use stochastic reverse-diffusion solvers (e.g., DDPM)? In that case, I suspect that some of the optimization difficulties should be attributed to stochasticity. Furthermore, recent techniques (e.g., DPM-Solver) can achieve impressive image quality in very few steps and that appears to be a better solution (both, theoretically and practically) to address vanishing gradients than residual connections.
>
> Yes, we use DDPM. Techniques such as DPM-Solver indeed offer advantages for reducing the number of diffusion steps and improving image quality, and we agree that exploring their application could be an interesting path for future research. Nonetheless, with the current focus of our work on dataset distillation, we believe that a broader analysis of diffusion model specifications, including DPM-Solver, is orthogonal to our primary contributions and extends beyond the intended scope of this paper. We hope to explore these promising directions in future work.
>
> > In addition, I feel it would be helpful to add a discussion on how hyperparameter T was set depending on the image resolution.
>
> We have added the used time steps in the corresponding sections for clarity.

---

> > ### Comment · Reviewer_9a7u · 2024-11-26
> >
> > Thank you for your experiments and responses. Some of my concerns are resolved. I have further questions:
> > 1. There are at least two intuitive methods to address the vanishing gradient problem you have described:
> > * Use an efficient sampling algorithm that achieves desired sample-quality in fewer steps: e.g., DPM-solver.
> > * Reduce number of steps during sampling: e.g., consider 3 steps (20,10,1) instead of 20 steps (20, 19, 18...1).
> >
> >   The solution proposed in this paper is to use the residual connections between timesteps. I feel that it is important to explain why these natural solutions are not preferable to address the vanishing gradient problem. Further, it may be possible that these methods are more effective than residual connections since these samplers do not alter the distribution of generated samples. Additionally, many works have backpropagated through the diffusion sampling: for an example, check DiffAttack[1]. Also see [2] for a memory-efficient backpropagation through diffusion sampling.
> >
> > 2. What generative prior is really being used in the data-distillation? First, the reverse-diffusion is altered with residual connections and the class-embeddings are also fine-tuned along with the latent-code.
> >
> > 3. How to set the hyperparameter T (10/20)?
> >
> > [1] Diffusion Models for Imperceptible and Transferable Adversarial Attack, TPAMI 2024.
> >
> > [2] AdjointDPM: Adjoint Sensitivity Method for Gradient Backpropagation of Diffusion Probabilistic Models, ICLR 2024.

---

> > > ### Author Response · Authors · 2024-11-26
> > >
> > > Thank you for your additional questions. We want to address these questions with the following responses:
> > >
> > > 1. We agree that our method opens new possibilities for optimizing the diffusion steps in diffusion models. We find further analysis along this axis an interesting avenue for future work. To summarize the goal of the current work: We wanted to integrate a pre-trained diffusion model that generates synthetic images, using the generated images as a bridge between diffusion models and distilled datasets. The computational graph, which connects the classification loss in the distilled dataset algorithm to the latent codes in diffusion models, contains several layers that can lead to the vanishing gradient problem. Furthermore, we want to highlight that the images in the distilled datasets do not need to be realistic like those used in diffusion models. With this in mind, we exploited residual connections similar to those used in ResNet or RNNs to facilitate the propagation of gradients. We found the inspiration derived from ResNets and RNNs to be more natural solutions.
> > > 2. We used DDPM (DDIM) from Stable Diffusion as a generative prior (as implemented by the initial work of LDMs by Rombach et al.).
> > > 3. We updated the work with this missing information for the respective experiments, essentially T=10.

---

> > > > ### Comment · Reviewer_9a7u · 2024-11-27
> > > >
> > > > Thank you for your response. However, this does not directly answer my questions.
> > > > * Q1. It is important to clarify why theoretically sound and natural solutions do not address the "vanishing gradients" in order to motivate the skip-connections between timesteps --- in my opinion, this is an important gap in the current version of the paper.
> > > > * Q2. The paper provides an argument in favor of generative prior but ends up using a completely different reverse-diffusion process and also alters the class-embeddings.
> > > > * Q3. The authors haven't reported *how* the hyperparameter T was adjusted.

---

> > > > > ### Author Response · Authors · 2024-11-28
> > > > >
> > > > > Thank you for clarifying the questions. Please find our responses:
> > > > >
> > > > > Q1. We agree that those approaches should be discussed in the current version of the paper, so we updated it accordingly. As mentioned earlier, the suggested approaches can also reduce the vanishing gradient problem. Yet, it should be noted that time step reduction approaches like DPM solver can be combined with our proposed solution and do not exclude it. LD3M is flexible regarding alternative timestep schedule schemes.
> > > > >
> > > > > Q2. We agree that our model uses a modified reverse-diffusion process and class embeddings due to the dataset distillation optimization. As the goal is to learn compressed and representative synthetic images, they do not necessarily have to follow the same data distribution as any real dataset. Additionally, distilled images are usually highly abstract, and standard class embeddings (e.g., obtained from a one-hot encoding scheme) are a strong constraint for these images. It should also be noted that truthful class embedding is not necessary for LD3M but helps, as shown with the FFHQ-pretrained LDM (Table 3). Also, altering class labels/embeddings (soft labels) is a common theme in dataset distillation, as evidenced by [1], [2] and [3].
> > > > >
> > > > > Q3. We updated the paper to explicitly mention that we selected the hyperparameter T based on the empirical evidence of performance/inference time trade-off (as shown in Fig. 5).
> > > > >
> > > > >
> > > > > References:
> > > > >
> > > > > [1] Yin et al. Squeeze, Recover and Relabel: Dataset Condensation at ImageNet Scale From A New Perspective. NeuRIPS 2023.
> > > > >
> > > > > [2] Yin et al. Dataset Distillation via Curriculum Data Synthesis in Large Data Era. TMLR 2024.
> > > > >
> > > > > [3] Lingao et al. Are Lare-scale Soft Labels Necessary for Large-scale Dataset Distillation? NeuRIPS 2024.

---

### Review · Reviewer_cHNx · 2024-11-12

**Summary Of Contributions:**

The paper studies dataset distillation and proposes LD3M (Latent Dataset Distillation with Diffusion Models), which integrates diffusion process into dataset distillation to address the challenges of distilling high-resolution images and enhancing generalization across different neural architectures. The experiments demonstrate its superior performance compared to GLaD (which uses StyleGAN-XL), and a decrease in computation resource overhead and distillation speed with configurable diffusion steps.

**Audience:**

Yes

**Claims And Evidence:**

Yes

**Requested Changes:**

Please see the weaknesses section.

Additionally, since LD3M utilizes pretrained LDM, I am curious whether using a pretrained generator could pose a problem when aiming to preserve the privacy of training data.

**Strengths And Weaknesses:**

### Strength
1. Dataset distillation is a critical area of research, especially for preserving training data privacy. There also remains a performance gap compared to full dataset fine-tuning, indicating substantial room for improvement.
2. The proposed method demonstrates superior performance across extensive experiments. With configurable diffusion steps, LD3M also allows a controlled balance between distillation speed and dataset quality.
3. The paper shows that off-the-shelf LDM can be directly integrated into LD3M.

### Weakness
1. While most experiments compare LD3M with GLaD, broadening the comparison to include a wider range of existing methods could provide a more comprehensive evaluation of LD3M's performance.
2. The paper mentions the preservation of training data privacy as an application of dataset distillation but does not detail how LD3M specifically contributes to enhancing privacy protections.
3. Although LD3M uses pretrained LDM and explores variations in pretraining data (as shown in Table 3), further investigation into how different pretraining datasets impact the quality of the distilled data would be beneficial. Could different pretraining data introduce biases into the distilled datasets?

---

> ### Author Response · Authors · 2024-11-21
>
> Thank you for your valuable comments, which we addressed in our updated submission. Please read the following responses to the mentioned weaknesses:
>
> 1. We agree that these experiments would be interesting. However, while most experiments compare LD3M with GLaD and various distillation algorithms (DC, DM, and MTT), this focus is intentional, as GLaD represents the state-of-the-art approach exploiting distillation in latent space with generative priors, making the most relevant baseline for our work. Comparing LD3M against GLaD allows us to directly highlight the advantages of incorporating diffusion models into this setting. Broader comparisons to other methods outside this generative prior framework, despite fascinating and potentially informative, will break the conditions and constraints of our problem.
> 2. We have expanded the introduction section to provide additional details on privacy applications, including a reference to emphasize this direction further.
> 3. Regarding potential biases from pre-trained generative priors, we demonstrated that LD3M produces more stable performances and images than GLaD, regardless of its prior training, even randomly initialized, as evidenced by Table 3 and visually in Figure 4. We agree that we can not guarantee a generation of distilled datasets that is more bias-free than GLaD. We believe the suggested experiments on more datasets for training the used LDM and exploring how training data affects biases in distilled datasets are fascinating avenues for future research. However, we find them orthogonal to the primary scope of this work, which is introducing diffusion models, which actually exploit the diffusion process, into dataset distillation with generative priors.
>
> > Additionally, since LD3M utilizes pretrained LDM, I am curious whether using a pretrained generator could pose a problem when aiming to preserve the privacy of training data.
>
> In alignment with the feedback from other reviewers, we have added an analysis of our modification in the context of classical image generation. This analysis demonstrates that our modification significantly changes the generation process, resulting in highly abstract images. This is a beneficial characteristic for dataset distillation, as dataset distillation does not require generating realistic images, such as in a classical image generation setting. Moreover, such abstraction reduces the likelihood of reproducing identifiable training data, thereby offering potential benefits for privacy preservation.

---

> > ### Comment · Reviewer_cHNx · 2024-11-26
> >
> > Thanks for the author's response. My concerns have been addressed.

---

### Decision · Action_Editor_vikb · 2024-12-13

**Recommendation:** Reject

**Comment:**

The reviewer scores for this paper are mixed. A primary concern highlighted revolves around the treatment of vanishing gradients with residual connections in reverse diffusion. To strengthen the paper, it is recommended that the authors include a comparative analysis with existing techniques known to address vanishing gradients. This would provide clarity on the necessity and potential advantages or limitations of the proposed method compared to these established approaches. The authors are encouraged to resubmit the manuscript after the major revision.

**Audience:**

The exploration of residual connections to mitigate vanishing gradients during reverse diffusion presents a new way that might appeal to researchers and practitioners working around diffusion models.

**Claims And Evidence:**

It seems that the claims made in the paper are not fully supported by convincing evidence. Reviewer point out that simpler, established methods such as reducing the number of diffusion steps could address the vanishing gradient issue, challenging the necessity of introducing residual connections. Furthermore, the paper's reliance on an existing method and a lack of extensive experimental validation suggest that the evidence presented may not adequately demonstrate the proposed solution's superiority or necessity over simpler alternatives.

**Resubmission Of Major Revision:**

The authors may consider submitting a major revision at a later time.